# Mechanistic studies of mycobacterial glycolipid biosynthesis by the mannosyltransferase PimE

Yaqi Liu [1], Chelsea M. Brown [2,3], Nuno Borges [4,5], Rodrigo N. Nobre[4], Satchal Erramilli [6], Meagan Belcher Dufrisne[1,7], Brian Kloss [1], Sabrina Giacometti [1,8], Ana M. Esteves[4], Cristina G. Timóteo[4], Piotr Tokarz[6], Rosemary J. Cater [1,9], Todd L. Lowary [10,11,12], Yasu S. Morita [13], Anthony A. Kossiakoff [6], Helena Santos [4]✉, Phillip J. Stansfeld [2]✉, Rie Nygaard [1,14]✉ & Filippo Mancia [1]✉

Tuberculosis (TB), a leading cause of death among infectious diseases globally, is caused by *Mycobacterium tuberculosis* (Mtb). The pathogenicity of Mtb is largely attributed to its complex cell envelope, which includes a class of glycolipids called phosphatidyl-*myo*-inositol mannosides (PIMs). These glycolipids maintain the integrity of the cell envelope, regulate permeability, and mediate host-pathogen interactions. PIMs comprise a phosphatidyl-*myo*-inositol core decorated with one to six mannose residues and up to four acyl chains. The mannosyltransferase PimE catalyzes the transfer of the fifth PIM mannose residue from a polyprenyl phosphate-mannose (PPM) donor. This step contributes to the proper assembly and function of the mycobacterial cell envelope; however, the structural basis for substrate recognition and the catalytic mechanism of PimE remain poorly understood. Here, we present the cryo-electron microscopy (cryo-EM) structures of PimE from *Mycobacterium abscessus* in its apo and product-bound form. The structures reveal a distinctive binding cavity that accommodates both donor and acceptor substrates/products. Key residues involved in substrate coordination and catalysis were identified and validated via in vitro assays and in vivo complementation, while molecular dynamics simulations delineated access pathways and binding dynamics. Our integrated approach provides comprehensive insights into PimE function and informs potential strategies for anti-TB therapeutics.

Tuberculosis (TB) is an ancient disease and[1,2] remains a significant global health challenge[3]. In 2023, it caused 1.25 million deaths and sickened 10.8 million people[3], surpassing COVID-19 as the deadliest infectious disease. The causative agent of TB is the pathogenic bacterium *Mycobacterium tuberculosis* (Mtb). What distinguishes mycobacteria pathogens from other bacteria is their cell envelope, which is a complex, multilayered structure that is hard to bypass and plays a crucial role in their survival and virulence[4–6]. The mycobacterial cell wall consists of an inner membrane rich in phospholipids and glycolipids, surrounded by a cell wall core composed of peptidoglycan covalently linked to arabinogalactan[7–10]. The cell wall core is esterified with long-chain mycolic acids, which form the inner leaflet of a unique

A full list of affiliations appears at the end of the paper. ✉e-mail: santos@itqb.unl.pt; Phillip.Stansfeld@warwick.ac.uk; rin7007@med.cornell.edu; fm123@cumc.columbia.edu

outer membrane structure often referred to as the mycomembrane[11–14]. The outer leaflet of this mycomembrane is populated with lipids and glycolipids, including phosphatidylinositol mannosides (PIMs), lipomannan (LM), lipoarabinomannan (LAM), trehalose monomycolates (TMM), and trehalose dimycolates (TDM), among others[15–18].

Among mycobacterial glycolipids of the inner membrane, PIMs are the most abundant[19–21]. They are characterized by the presence of one to six mannose residues and up to four acyl chains attached to their phosphatidyl-*myo*-inositol (PI) anchor[18,19]. PIMs are important components of the mycobacterial cell envelope, contributing to its structural integrity, regulating permeability, and mediating host–pathogen interactions[17,18,22,23]. Moreover, some PIMs serve as precursors for lipomannan (LM) and lipoarabinomannan (LAM), two abundant large lipoglycans in the mycobacterial cell envelope[24]. These hyperglycosylated derivatives are characterized by a linear mannan core, with LAM additionally containing arabinose branches[18,24].

The early steps of the PIMs biosynthetic pathway occur on the cytoplasmic side of the inner membrane and have been partially characterized at both genetic and biochemical levels[19,20]. The mannosyltransferase PimA (Rv2610c) and the mannosyltransferase PimB (Rv2188c) transfer mannose residues from GDP-mannose, to the C2 and C6 hydroxyl groups of the *myo*-inositol moiety of PI, respectively, yielding PIM1 and PIM2[25–27] (Fig. 1a). The acyltransferase PatA (Rv2611c) then uses palmitoyl-CoA as a donor to acylate the C6 hydroxyl group of the mannose ring linked to the 2-OH position of the inositol in PIM1 and PIM2, resulting in AcPIM1 and AcPIM2, respectively. An additional acyl chain moiety can be added to the C3 hydroxyl group of the *myo*-inositol of AcPIM2 to form $Ac_2PIM2$, but the enzyme carrying out this reaction remains unknown[28]. These acylated intermediates are presumed to translocate to the outer leaflet of the membrane, a process that is likely to require a currently uncharacterized flippase. Further mannosylation of $Ac_{1/2}PIM2$ to $Ac_{1/2}PIM3$ has been attributed to PimC, although the gene is only present in some strains of Mtb, suggesting that there is another unidentified gene that mediates the reaction[29,30]. The subsequent mannosylation of $Ac_{1/2}PIM3$ to $Ac_{1/2}PIM4$ is again presumed to be catalyzed by an unidentified enzyme denoted as PimD (Fig. 1a)[29,31]. The transfer of the fifth mannose to form $Ac_{1/2}PIM5$ is known to be catalyzed by PimE[30]. Finally, the enzyme responsible for the final step of converting $Ac_{1/2}PIM5$ to $Ac_{1/2}PIM6$ remains unknown (Fig. 1a). A putative enzyme previously named PimF was initially thought to be involved in high molecular weight PIMs[32]; however, subsequent research has shown that this enzyme, now renamed LosA, is involved in the biosynthesis of an unrelated family of glycosylated acyltrehalose lipooligosaccharides in *M. marinum*[33].

PimE, the focus of our studies, is a polyprenyl-monophospho-β-D-mannose (PPM)-dependent mannosyltransferase that catalyzes the transfer of the fifth mannose from a PPM donor to the $Ac_1PIM4$ acceptor substrate, leading to the formation of $Ac_1PIM5$ via an $α(1 → 2)$ glycosidic bond (Fig. 1a, b)[30,34,35,36]. PPM is synthesized by PPM synthase (Ppm1) using GDP-mannose and polyprenyl phosphate (PP) as precursors[37]. The length of the polyprenyl chain can vary, with the two most common forms being decaprenol (C50) and nonaprenol (C35)[35,38]. Previous studies have investigated the functional importance of specific residues in PimE, and D58 in *M. smegmatis* PimE, was found to be essential for enzymatic activity[30]. PimE has been identified as an intrinsic resistance determinant, affecting sensitivity to multiple antibiotics[39]. PimE-deletion strains exhibit defective planktonic growth and aberrant morphology[39,40]. These mutants also showed increased sensitivity to multiple antibiotics, including erythromycin, vancomycin, and cefotaxime, along with increased copper sensitivity and compromised envelope permeability[30,39–41]. The genetic ablation of PimE leads to the accumulation of $Ac_{1/2}PIM4$ and a deficiency in the synthesis of $Ac_{1/2}PIM6$, which has been shown to have significant consequences for the structural integrity of the mycobacterial cell envelope and plasma membrane[30,41].

While these findings underscore the relevance of PimE in the biosynthesis of PIMs and its role in maintaining the mycobacterial cell envelope integrity, the structural basis for substrate recognition and catalysis by this enzyme remains elusive. In this study, we used single-particle cryo-electron microscopy (cryo-EM) to determine high-resolution structures of PimE in the apo form and in a complex bound with its mannosylated product $Ac_1PIM5$ and by-product PP at 3.0 and 3.5 Å resolution, respectively. Our structural analysis, complemented by molecular dynamics (MD) simulations and in vitro and in vivo functional assays, provides insights into the catalytic mechanism of PimE and the structural basis for substrate recognition.

## Results

### Structure determination of apo PimE

To identify a suitable candidate for structural studies, we screened PimE orthologs from 15 mycobacterial species, assessing their expression levels in *E. coli*, and solubility and stability in detergent. PimE from *Mycobacterium abscessus* (*Ma*PimE) exhibited the most promising properties and was therefore selected for further characterization. To confirm that *Ma*PimE retained enzymatic activity, we expressed it in *E. coli*, and incubated the isolated membrane fraction with its two substrates, $Ac_1PIM4$ and PPM (see Materials and Methods for substrate preparation). The reaction products were analyzed by thin-layer chromatography (TLC), demonstrating the catalytic activity of recombinantly-expressed *Ma*PimE (Supplementary Fig. 1a).

For structure determination, *Ma*PimE was purified in *n*-dodecyl-β-D-maltoside (DDM) and reconstituted into lipid-filled nanodiscs (Supplementary Fig. 1c, d). *Ma*PimE has a molecular weight of 46 kDa, which poses a challenge for structure determination by cryo-EM[42,43]. To overcome this hurdle, we screened a synthetic phage display library to identify recombinant antigen-binding fragments (Fabs) that could specifically bind to *Ma*PimE, to increase the size of the particles and add an extra-membrane feature to facilitate their alignment for cryo-EM data processing[42,44,45]. We evaluated eight Fab candidates for their ability to form stable complexes with *Ma*PimE, and we selected Fab-E6 due to its high binding affinity (Supplementary Fig. 1e, f).

We collected 7469 micrographs of nanodisc-reconstituted apo *Ma*PimE in complex with Fab-E6 on a Titan Krios transmission microscope equipped with a K3 direct electron detector. After iterative 2D classification, we obtained high-quality 2D class averages showing bound Fab and secondary structural features within the nanodisc-embedded transmembrane (TM) domain. After further particle sorting and 3D refinement, we obtained a map with an overall resolution of 3.0 Å (Fig. 1c, Supplementary Table 1 and Supplementary Figs. 2a–c, 3a–c).

### Structure of PimE

We were able to construct an almost complete atomic model of PimE with the exception of 20 residues at the N-terminus and ten residues at the C-terminus, and three disordered loop regions (80–87, 246–250, and 369–376) (Fig. 1d–f and Supplementary Fig. 4a). PimE contains twelve TM helices with both the N- and C-termini located on the cytoplasmic side of the membrane. These TM helices vary in length, ranging from 11 to 27 amino acids, with TM helix 12 being the longest (27 amino acids) and TM helix 6 the shortest (11 amino acids), spanning approximately two-thirds across the membrane (Figs. 1d–f, 2a). The TM helices are interconnected by five short cytoplasmic loops (CL1–CL5), three periplasmic loops (PL1–PL3), three membrane-embedded loops, and four juxtamembrane (JM) helices. Additionally, there are smaller connecting segments within these loops, particularly between the JM and the TM helices, designated interconnecting loops (IL). PL1, located between TM helices 1 and 2, contains two JM helices,

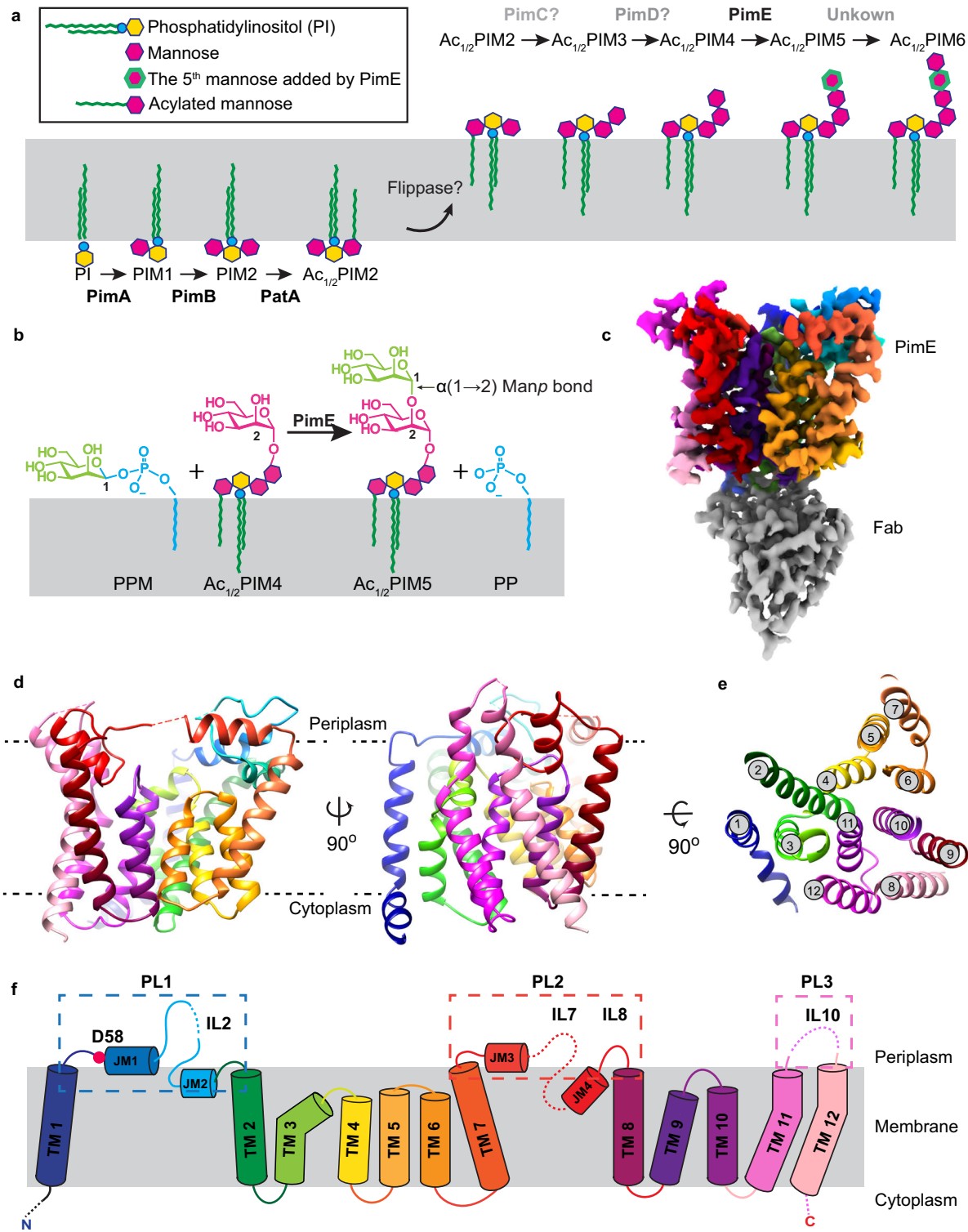

**Fig. 1 | Biosynthetic pathway and structural architecture of PimE. a** Overview of PIM biosynthesis in mycobacteria, featuring the enzymatic reactions carried out by key mannosyltransferases. Enzyme names are indicated above the reaction arrows. The early steps occur on the cytoplasmic side of the plasma membrane, where PimA and PimB transfer mannose residues from GDP-mannose to the 2-OH and 6-OH positions of the inositol moiety of phosphatidylinositol (PI), yielding PIM1 and PIM2, respectively. PatA then acylates the 6-OH position of the mannose ring of PIM2, resulting in AcPIM2. An unknown acyltransferase can add an additional acyl chain to the 3-OH of the inositol of AcPIM2 to form Ac₂PIM2. These acylated mannosylated intermediates are presumed to translocate to the outer leaflet of the membrane by an unknown flippase. PimC and PimD are thought to catalyze the subsequent tri- and tetra-mannosylation steps, leading to the formation of Ac₁PIM4.

PimE catalyzes the transfer of the fifth mannose residue from polyprenyl-mono-phospho-β-D-mannose (PPM) to Ac₁PIM4, forming Ac₁PIM5. The enzyme responsible for the conversion of Ac₁PIM5 to AcPIM6 remains unknown. **b** Mannosyl transfer reaction catalyzed by PimE, showing the formation of an α(1 → 2) glycosidic bond between PPM and Ac₁PIM4, yielding Ac₁PIM5 and PP. Only tri-acylated PIMs (Ac₁PIM4 and Ac₁PIM5) are shown for the sake of simplicity. **c** Cryo-EM map of *Ma*PimE in complex with Fab-E6. PimE is depicted in rainbow, while the Fab-E6 is shown in gray. (d) PimE is shown as a ribbon colored in rainbow, as in Fig. 1f. **e** TM helix arrangement of PimE is depicted as a cross-section colored in a rainbow as in Fig. 1f. **f** Topological diagram of PimE showing the arrangement of TM segments and extracellular domains. The key catalytic residue D58 is shown as a red dot.

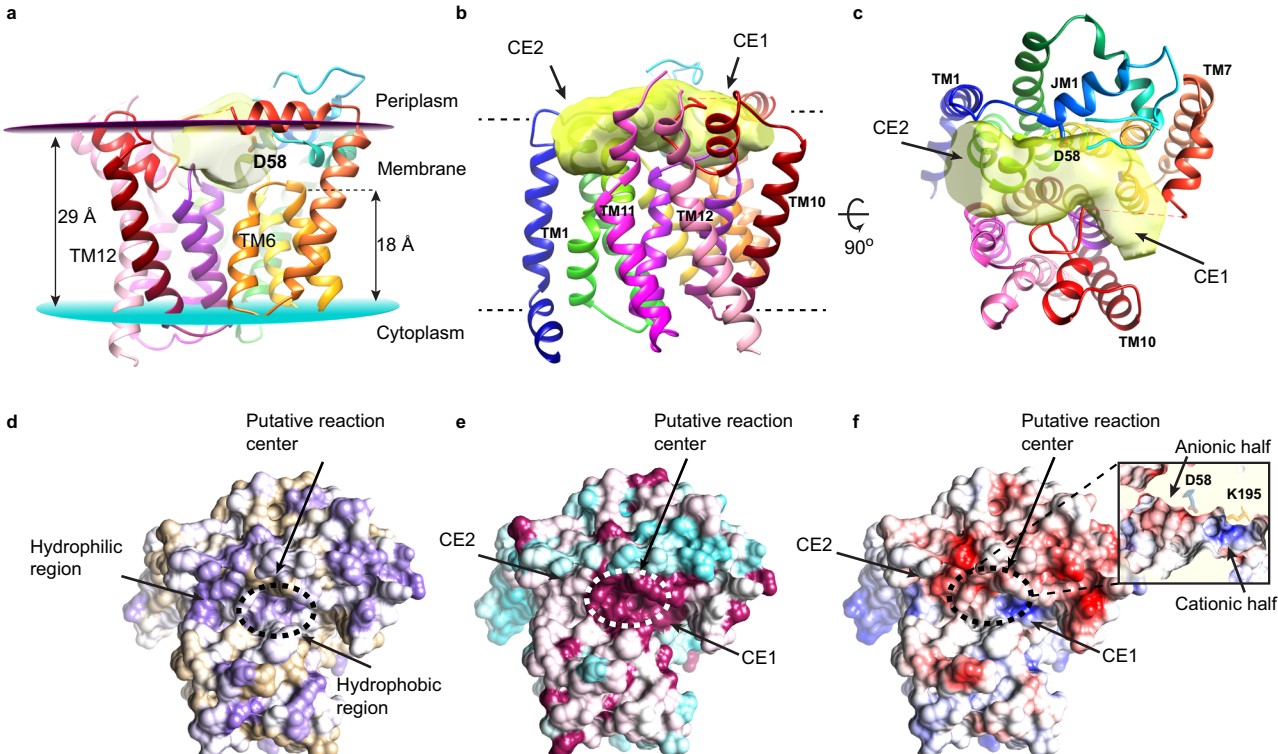

**Fig. 2 | The putative substrate-binding cavity of PimE. a** The putative active site cavity within PimE is colored in semitransparent yellow, showing its elongated, cashew-like shape and orientation relative to the membrane plane. **b, c** PimE viewed parallel (**b**) and perpendicular (**c**) to the membrane plane, with the elongated, cashew-shaped cavity shown in yellow. The cavity is oriented almost parallel to the membrane plane, with its two ends (Cavity End 1, CE1, and Cavity End 2, CE2) curving slightly toward the membrane. The cavity is surrounded by periplasmic loops (PL1 to PL3) and the connecting loops between TM helices 3 and 4 and TM helices 9 and 10. **d–f** PimE rendered in surface representation colored by hydrophobicity on a purple (very hydrophilic) to tan (very hydrophobic) scale (**d**) by conservation on a cyan (low conservation) to magenta (absolute conservation) scale (**e**) and by electrostatic potential (**f**) on a range of ±5 kBT/e. The putative reaction center at the central part of the cavity is marked with a dotted circle.

JM1 and JM2, connected by a loop (IL2). JM1 is amphipathic and lies at the interface between the periplasmic space and the TM domain, parallel to the membrane plane (Figs. 1d, e, 2a). JM2 is positioned right below the interface, almost parallel to the membrane plane, and buried inside the membrane. PL2, which connects TM helices 7 and 8, also features two JM helices, JM3 and JM4, connected by a loop (IL7) (Fig. 1d–f). JM3 is situated at the periplasmic-membrane interface, while JM4 adopts an unusual orientation, being buried inside the membrane at an angle of approximately 60° relative to the plane of the membrane.

The variable domain of the Fab was well resolved, allowing for reliable modeling. In contrast, the constant domain of the Fab was not included in the final model, due to the lack of well-defined density (Fig. 1f and Supplementary Fig. 1g). The cryo-EM map reveals well-resolved density at the interface between the Fab-E6 heavy chain and the cytoplasmic loops of *Ma*PimE (Supplementary Fig. 1g). Only the heavy chain appears to contribute to the interaction interface, which is stabilized through hydrogen bonds between its complementarity-determining regions (CDRs) and the cytoplasmic loops that connect TM helices 2 and 3, TM helices 4 and 5, and TM helices 8 and 9 in PimE (Supplementary Fig. 1g).

**Structural homology of PimE**

Using the structural homology DALI server[46] we confirmed that PimE belongs to the GT-C superfamily of glycosyltransferases. This superfamily can be further classified into the GT-C$_A$ and GT-C$_B$ subclasses[47,48]. The enzymes with the highest structural homology to PimE include yeast glucosyltransferase ALG6 (PDB: 6SNH)[49], bacterial arabinosyltransferase ArnT (PDB: 5F15)[50], and bacterial oligosaccharyltransferase PglB[51] (PDB: 5OGL) (Supplementary Fig. 5a–d), all belonging to the GT-C$_A$ subclass[47]. These enzymes share a conserved structural motif comprising, for PimE, the first seven TM helices (TM helices 1–7) and JM1 and JM2. Notably, D58 in *Ma*PimE, which corresponds to the previously identified catalytic residue D58 in *M. smegmatis* PimE[30] (Supplementary Fig. 5a–d), is located at the apex of JM1. This position is equivalent to the catalytic aspartate residues in ALG6, ArnT, and PglB, further supporting its role as the essential catalytic base (Supplementary Fig. 5a–d). The DALI search also identified mycobacterial arabinosyltransferase AftA (PDB: 8IF8)[52] as a structural homolog to PimE. While not explicitly classified in earlier reviews due to the recent publication of its structure, AftA likely also belongs to the GT-C$_A$ subclass. Interestingly, the search also revealed structural similarities to proteins outside the GT-C family, such as PIGU, a subunit of the human GPI transamidase complex[53] (PDB: 7WLD) (Supplementary Fig. 5e, f). Despite the conserved fold, PimE and PIGU differ considerably in their biological functions and the reaction they catalyze: PimE catalyzes a glycosyl transfer reaction in the biosynthesis of PIMs using polyprenyl-phosphomannose as a donor substrate, whereas PIGU attaches GPI anchors to proteins in eukaryotic cells. Their active site composition also differs significantly.

To assess the structural relevance of PimE across mycobacterial species, we compared the *M. abscessus* PimE structure with AlphaFold-predicted models of PimE from *M. tuberculosis* and *M. smegmatis* (Supplementary Fig. 6). We chose these two species as *M. tuberculosis* is a clinically significant pathogen, while *M. smegmatis* is an established model organism for mycobacterial research. The predicted models showed high structural similarity to *M. abscessus* PimE, with an amino acid sequence identity of 64 and 61%, and an RMSD of 2.5 and 1.9 Å,

across 369 Cα pairs for *M. tuberculosis* PimE and *M. smegmatis* PimE, respectively. Such conservation demonstrates that our structures may serve as an appropriate framework for studying PimE function in pathogenic mycobacteria and may help guide future therapeutic strategies.

## Putative substrate-binding cavity of PimE

We identified a prominent cavity within PimE, located at the interface between the TM domain and the periplasm. This elongated cavity adopts a cashew-like shape and is oriented almost parallel to the membrane plane, with its two ends curving slightly towards the membrane (Fig. 2a–c). The cavity is surrounded by three periplasmic loops (PL1, 2, and 3) and the connecting loops IL7 and IL9 (Figs. 1f, 2a–c). For clarity, we refer to the two ends of the cavity as "Cavity End 1" (CE1) and "Cavity End 2" (CE2) (Fig. 2b, c). CE1 is defined by JM3 and TM helices 5, 6, and 9, while CE2 is defined by JM1 and JM2 (Fig. 2b, c).

Given its location and structural features, we hypothesized that this cavity serves as the substrate-binding site of PimE. The central part of the cavity, situated near the membrane–periplasm interface, is largely hydrophilic and highly conserved (Fig. 2d–e and Supplementary Fig. 7) and contains the predicted catalytic residue D58 (Fig. 2c). In contrast, the region of the cavity proximal to CE1 is predominantly hydrophobic, while the region on the CE2 side is more hydrophilic (Fig. 2d and Supplementary Fig. 8b). Furthermore, the electrostatic surface of the cavity reveals an uneven charge distribution, with the central region divided into two halves, a cationic half pointing towards CE1 and an anionic one towards CE2 (Fig. 2f and Supplementary Fig. 8c). The positive charge in the cationic half is partly caused by K195 which is highly conserved (Fig. 2f and Supplementary Figs. 7, 8c), while the anionic nature of the other half is mainly due to the presence of D58 (Fig. 2a–c). These properties suggest that the central region of the cavity serves as the reaction center (Fig. 2d–f), bringing together the hydrophilic head groups of the donor and acceptor substrates (Supplementary Fig. 8). We further hypothesized that CE1 acts as the entry point for the donor substrate PPM, with the hydrophobic region proximal to CE1 accommodating its polyprenyl tail. Conversely, CE2 could serve as the entry point for the acceptor $Ac_{1/2}PIM4$, with its extended hydrophilic sugar head positioned along the cavity from the center towards CE2 and the acyl chain of $Ac_{1/2}PIM4$ extending towards the TM domain.

## The cryo-EM structure of reaction products-bound *Ma*PimE

To validate our hypotheses and gain a deeper mechanistic understanding of this enzyme, we set forth to determine the structure of substrate-bound PimE. To this end, we added $Ac_1PIM4$, isolated from the membranes of *M. smegmatis* $mc^2155$ Δ*pimE*[54], and PPM−enzymatically derived from *E. coli* BL21 (DE3) PLysS cells hosting a PPM synthase gene from *M. tuberculosis* H37Rv[55]−during the nanodisc incorporation stage of *Ma*PimE purification. We added Fab-E6 to this complex and determined its cryo-EM structure to 3.5 Å resolution following a similar flowchart to the one used for the apo structure (Supplementary Table 1 and Supplementary Figs. 2d–f, 3d–f).

The resulting structure revealed an overall topology comparable to the apo structure (Fig. 3a, b). However, we observed two prominent and interpretable densities in the central cavity. One, close to CE1, was consistent with the phosphate group and the first three prenyl units of PPM. The absence of any density where we could fit the mannose moiety led us to interpret it as the PP portion of the PPM donor (Fig. 3 and Supplementary Fig. 9c). On the opposite side of the cavity, near CE2, the second density shows key characteristics of $Ac_1PIM5$, and we could fit the inositol ring, five mannose residues, the phosphate group, and the first two carbons of the glycerol backbone directly connected to the phosphate group into this density (Fig. 3c–h and Supplementary Fig. 9c). However, the third carbon of the glycerol backbone and the acyl chains extending from it could not be traced reliably in the density map. Similarly, the acyl chain bound to the C6 hydroxyl group of the

mannose residue linked to the O2 of the inositol remained elusive beyond the carbonyl group directly attached to the mannose (Fig. 3). The weaker density for the mannose and phosphate group furthest from the reaction center likely reflects flexibility in these distal parts of the molecule. The presence of PP and $Ac_1PIM5$ in the structure, the products of the PimE reaction, despite having added PPM and $Ac_1PIM4$, suggests that the mannosyl transfer reaction has occurred.

Superposition of the apo and product-bound structures revealed close structural similarity (Cα RMSD of 1.5 Å across all 367 Cα pairs) with subtle conformational rearrangements. (Supplementary Fig. 8d). The TM domain of the product-bound structure exhibited minimal deviations from the apo structure, with a modest inward pivot rotation/translation of TM helix 7 being the most notable change. More noticeable rearrangements were observed in the PLs (Supplementary Fig. 8d). In PL1, the flexible region within IL2, connecting JM1 and JM2, which was previously disordered and unresolved in the apo structure, became traceable in the product-bound complex (Fig. 3c, f–h and Supplementary Fig. 8d). Similarly, the extended loop (IL7) between JM3 and JM4 in PL2 was disordered in the apo form, but in the product-bound complex, IL7 became well-resolved, and an additional small helix (JM') was formed (Fig. 3c, f and Supplementary Fig. 8d). This ordered IL7 appears to form an "arch" which the substrate threads its way under and into the active site cavity (Fig. 3c, f and Supplementary Figs. 5a, 8a). PL3, which connects TM helices 11 and 12, also transitioned from a partially disordered state in the apo form (Fig. 3c–f and Supplementary Fig. 8d) to a fully ordered and traceable conformation in the product-bound structure (Fig. 3c–f and Supplementary Fig. 8d). These subtle structural differences, in particular the ordering of specific loops and the formation of defined substrate pathways, suggest localized adjustments that could facilitate substrate positioning and stabilization during PimE's catalytic cycle.

Both PP and $Ac_1PIM5$ adopt a curved shape, with their polar head groups projecting into the hydrophilic reaction center of the substrate-binding cavity (Fig. 3 and Supplementary Fig. 8a). The phosphate group of PP is located along the cationic half of the cavity (Supplementary Fig. 8c). At the same time, the mannose head of $Ac_1PIM5$ is positioned along the anionic half (Supplementary Fig. 8c). Only the first three prenyl units of the polyprenyl chain of PP could be resolved, with the chain pointing towards the hydrophobic groove between TM helices 6 and 9 (Fig. 3f and Supplementary Fig. 9c). Based on the observed orientation and the hydrophobic nature of the groove, it is likely that the remaining prenyl units of the polyprenyl chain extend into this region (Fig. 3f). On the opposite side, $Ac_1PIM5$ is surrounded by PL1, JM1, and JM4. While the acyl chains of $Ac_1PIM5$ could not be fully traced in the map, their overall spatial arrangement suggests that they probably extend toward the membrane region near TM helix 3 (Fig. 3e, h).

## Structural and computational insights into ligand interactions

Our structure reveals a network of interactions between PimE and its ligands. At the center of the binding pocket, D58, located at the tip of JM1, forms a hydrogen bond with the C2 hydroxyl group of the fifth mannose of $Ac_1PIM5$. Y62 forms a hydrogen bond with D58, potentially stabilizing its orientation within the active site (Fig. 4a). H321 engages with the fifth mannose moiety via a hydrogen bond to the C6 hydroxyl group. Y161 interacts with the glycosidic oxygen linking the third and fourth mannose residues through a hydrogen bond. D55 coordinates the third mannose residue via a hydrogen bond with its C4 hydroxyl group. W361 forms a hydrogen bond with the C3 hydroxyl group of the inositol moiety in $Ac_1PIM5$, D160 coordinates with the inositol center through a hydrogen bond to the C2 hydroxyl group, and R43 establishes a hydrogen bond with the C3 hydroxyl group of the first mannose residue (Fig. 4a, b).

The phosphate group of PP is coordinated by three conserved residues, H322, W319, and K195 (Fig. 4a). K195 forms a salt bridge with

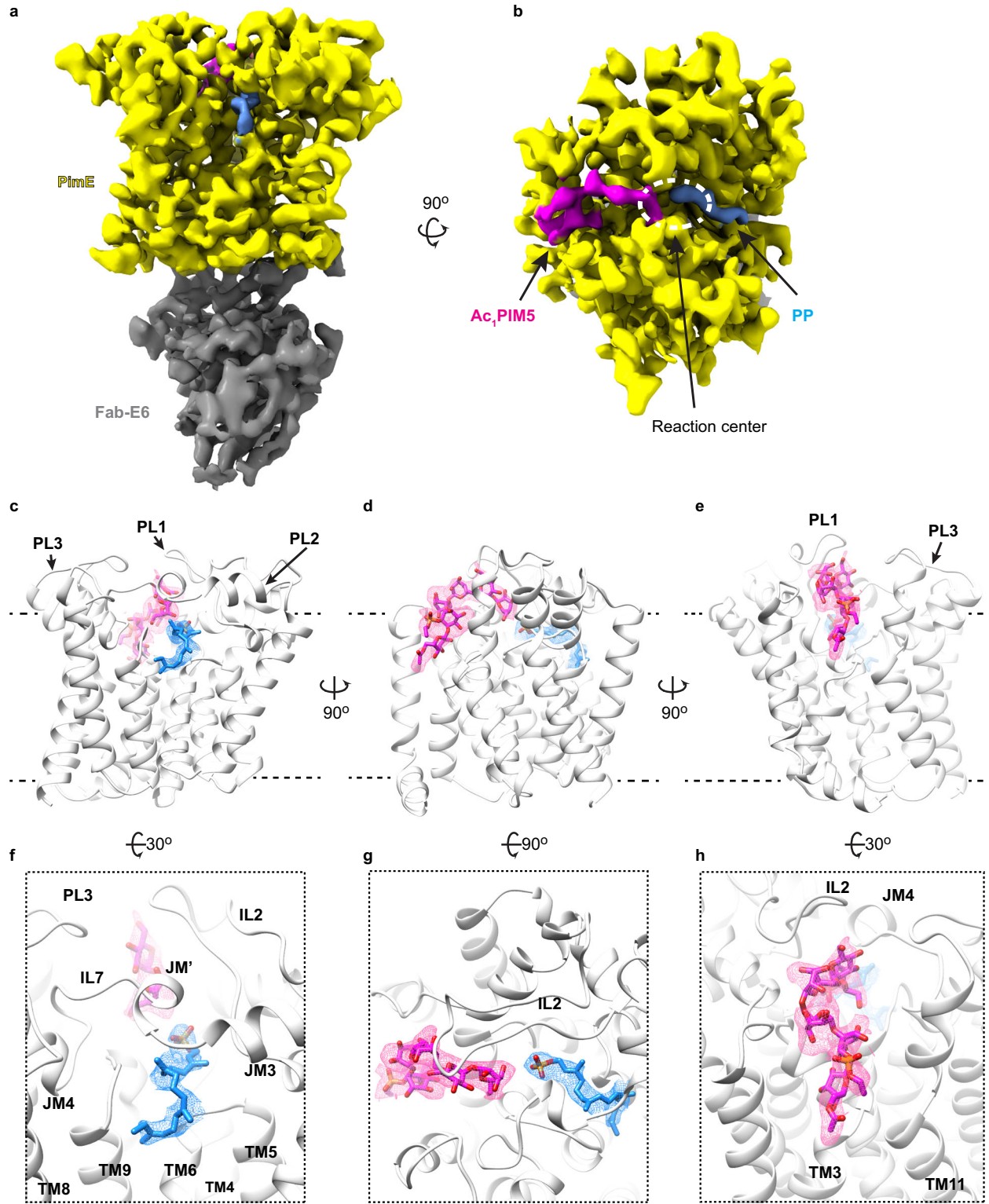

**Fig. 3 | The cryo-EM structure of substrate-bound *Ma*PimE. a, b** Cryo-EM density for the product-bound complex of *Ma*PimE with Ac$_1$PIM5 (product) and PP (by-product), viewed parallel to the membrane plane. Fab-E6 binds to the same cytoplasmic domain of *Ma*PimE as observed in the apo structure. **c–e** Ribbon representation for the product-bound structure of PimE (white) bound with PP (blue) and Ac$_1$PIM5 (magenta) shown in different orientations. PP and Ac$_1$PIM5 adopt a curved shape, with their polar head groups projecting into the hydrophilic center of the substrate-binding cavity. **f–h** A focused view of the region where PP and Ac$_1$PIM5 bind. PP and Ac$_1$PIM5 adopt a curved shape. The polyprenyl chain of PP points towards the hydrophobic TM domain groove composed of TM helices 6 and 9, while Ac$_1$PIM5 is surrounded by PL1, JM1, and JM4, with its acyl chains likely extending toward the TM region near TM helix 3.

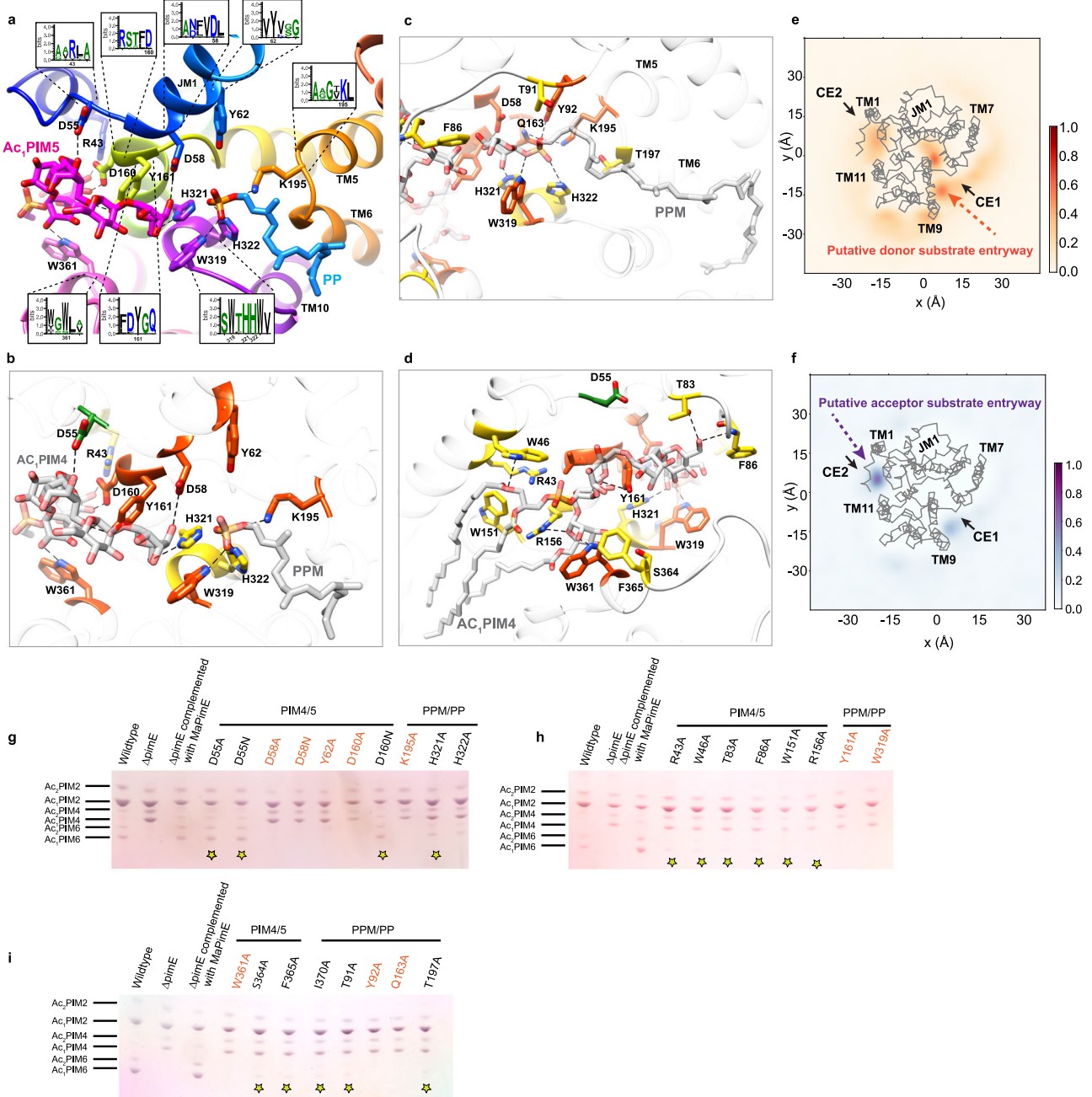

**Fig. 4 | Structural insights into *Ma*PimE: Active site, substrate interactions, and functional residues. a** Close-up view of the active site from the cryo-EM product-bound structure of PimE bound with products Ac₁PIM5 and by-product PP. Key residues are shown as sticks. Insets: Sequence logos highlighting the conservation of active site residues involved in interactions with PP or Ac₁PIM5. **b** Same view as (**a**) with residues colored according to mutational effects: red-orange for complete activity loss, yellow for reduced activity, and green for no change in activity. **c**, **d** RoseTTAFold docked models of PimE with full-length donor PPM and acceptor Ac₁PIM4. Residues are colored as in (**b**) based on mutational effects. Hydrogen bonds are shown as black dotted lines in panels (**a**–**d**). **e**, **f** Density of PPM (**e**) and Ac₁PIM4 (**f**) in CG-MD simulations with respect to PimE, with the protein backbone shown as gray lines. Regions of darker color show higher density by the lipid. Key areas of the protein are highlighted. **g**–**i** HPTLC analysis of PIMs profiles from *M. smegmatis* strains. Each plate compares PIMs from WT *M. smegmatis* mc²155, Δ*pimE* mutant strain, and Δ*pimE* complemented with WT or mutant *Ma*PimE. Major PIM species are indicated. Yellow stars mark mutants with reduced but not abolished activity (detectable PIM6 production). Mutations causing complete loss of activity are labeled in red-orange. All experiments were independently repeated two times with similar results.

the phosphate, while H322 forms a hydrogen bond with one of the phosphate oxygens. W319 provides additional stabilization through a cation-π interaction with the positively charged K195 (Fig. 4a).

To complement our structural data and explore interactions with full-length substrates, we employed a combination of molecular docking and RoseTTAFold[55] modeling. The RoseTTAFold models showed good agreement with the experimentally observed

interactions in the catalytic site and provided additional insight into potential interactions involving the regions of the polyprenyl chain of PPM and the acyl chains of Ac₁PIM4/5.

For the donor substrate, the RoseTTAFold models support the hypothesis that H322, W319, and K195 coordinate the phosphate group of PPM, consistent with our observations from the cryo-EM structure. Additionally, the models suggest that Y92 and Q163 contribute to this

coordination network through hydrogen bonding with the phosphate group (Fig. 4c). H321 appears to interact with the mannose head group of PPM (Fig. 4c), complementing its interaction with the fifth sugar moiety of Ac$_1$PIM5, as observed in our cryo-EM structure (Fig. 4a). The models also show that the decaprenyl tail of PPM fits well into the hydrophobic groove, with the unresolved prenyl units extending along TM6, again, consistent with our structural observations.

With respect to the acceptor substrate, the RoseTTAFold model also provides insights into interactions not fully captured in the cryo-EM structure. The model suggests that Y161 may coordinate the inositol moiety via a hydrogen bond, while T83 and F86 coordinate the fourth mannose residue (Fig. 4d). R156 is shown to coordinate the first mannose residue. W46 is predicted to coordinate the ester oxygen that connects the sn-2 acyl chain to the glycerol backbone of the phosphatidylinositol in Ac$_1$PIM5 (Fig. 4d). Our models also indicate the presence of several amino acid residues in the vicinity of the substrate-binding sites, including W151, F365, S364, T91, and T197, which may contribute to substrate recognition, orientation, or active site organization (Fig. 4c, d).

To further explore the dynamics of the cavity and the potential entry points for the substrates, we performed coarse-grained molecular dynamics (CG-MD) simulations. The simulations showed that the hydrophobic region proximal to CE1 favors the binding of PPM, with its lipid tail extending towards the TM domain (Fig. 4e). In contrast, the more hydrophilic region proximal to CE2 preferentially accommodates Ac$_1$PIM4, with its sugar head positioned along the cavity from the center towards CE2 (Fig. 4f). These findings align with our structural analysis and provide additional insights into the substrate binding and entry mechanisms.

In addition to the CG-MD studies, simulations at atomistic resolution were performed in triplicate for both the substrates and products of glycosyltransferase reaction to obtain molecular-level insights into the interactions. The ligands remained bound throughout both sets of 500 ns simulations (Supplementary Fig. 10a), indicating the stability of the bound ligands in these positions. This can also be seen by the defined contact areas (Supplementary Fig. 10b–d) for both substrates and products. The increased stability of regions in the protein with the addition of the ligands seen in the resolved structures are also observed in the movement of the protein (Supplementary Fig. 11a, b) with and without ligands. The regions between JM1 and JM2 and the region between JM4 and JM5 are more dynamic when the binding partners are not present in the simulations. The simulations can provide further evidence to the key residues involved, with the $pK_a$ of residues D58 and K195 fluctuating in different conditions (Supplementary Fig. 11c), suggesting a possible role in catalysis through their predicted changes in protonation state.

## Functional characterization of structure-based PimE mutants

To validate our structural and computational results, we designed specific mutations and performed in vivo complementation assays by expressing wild-type (WT) and mutant versions of *Ma*PimE in *M. smegmatis* Δ*pimE* and analyzed their PIMs profiles using thin-layer chromatography (TLC). As expected, the D58A and D58N mutations resulted in a complete loss of activity (Fig. 4g). Nearby, Y62 forms a hydrogen bond with D58, and again not surprisingly, the Y62A mutation also led to a complete loss of activity (Fig. 4g). Mutations affecting residues involved in coordinating the phosphate group of PPM and/or PP showed varying effects. K195A, Y92A, Q163A, and W319A mutations resulted in a complete loss of activity (Fig. 4g). In contrast, H321A and H322A mutants showed reduced activity, evidenced by some accumulation of Ac$_1$PIM4 and detectable, albeit diminished, production of AcPIM6 (Fig. 4g).

We also mutated several residues interacting with the acceptor substrate. D160A led to drastically reduced activity, while D160N showed PIMs profiles comparable to the WT enzyme, suggesting that

hydrogen bonding capability at this position is sufficient for function (Fig. 4g). Alanine mutations of W361 and Y161 resulted in a complete loss of activity (Fig. 4h–i), whereas D55A and D55N mutations did not affect activity, suggesting a non-essential role for this residue (Fig. 4g). R43A, T83A, F86A, and R156A all showed accumulation of Ac$_1$PIM4 yet detectable production of AcPIM6, indicating reduced but not abolished activity (Fig. 4h).

Furthermore, mutations of residues in the vicinity of the substrate-binding sites (W151, F365, S364, T91, and T197) all showed accumulation of Ac$_1$PIM4 yet detectable production of AcPIM6, indicating again reduced but not abolished activity (Fig. 4h–i). This suggests that they contribute to efficient catalysis without being essential for the reaction.

To complement the functional data described above, we performed in vitro enzymatic assays using membrane fractions isolated from *E. coli* expressing WT and mutant PimE. Here, purified Ac$_1$PIM4 and enzymatically synthesized PP[U-$^{14}$C]M were added to the membrane fractions, and the formation of $^{14}$C-Ac$_1$PIM5 was monitored by TLC. Consistent with our in vivo results (Fig. 4g, h), D58A, D58N, K195A, W319A, and Y62A mutations completely abolished activity (Supplementary Fig. 9d, e). H321A and H322A showed reduced but not abolished activity (Supplementary Fig. 9d, e). D55A and D55N mutations did not significantly affect activity (Supplementary Fig. 9d, e). The D160A mutation largely reduced PimE activity, while D160N only partially reduced it (Supplementary Fig. 9d, e).

All residues identified as important for catalytic function are strictly conserved across mycobacterial PimE homologs (Supplementary Figs. 6, 7), these include the catalytic residue D58 and residues whose mutations resulted in substantial loss of enzymatic activity in our functional assays, supporting the broader relevance of our findings for understanding PimE function in pathogenic mycobacteria.

## Mechanism of substrate entry and catalysis

Metal ions play crucial roles in the catalytic activity of many glycosyltransferases[56]. However, the metal dependency of enzymes involved in PIMs biosynthesis, including PimE, remains poorly characterized. To investigate the potential metal dependency of PimE, we performed activity assays under various conditions. PimE remained fully active in the presence of metal chelators (EDTA and EGTA) (Supplementary Fig. 1b, left). Furthermore, the addition of Mg$^{2+}$, a common cofactor for glycosyltransferases, similarly showed no detectable effect on product formation (Supplementary Fig. 1b, right). These results suggest that PimE functions independently of metal ions. This metal-independent behavior of PimE aligns with the absence in its sequence of a conserved DxD motif, which is characteristic of metal-dependent glycosyltransferases.

This observation, together with our analysis of the product-bound structure and insights from the MD simulations, allows us to propose a catalytic mechanism for PimE. We suggest that the donor substrate PPM accesses the active site cavity through an arch formed by IL7 (Fig. 5a). Once in the active site, the polar head of PPM is positioned with its phosphate group coordinated by the positively charged side chains of K195 and H322 (Fig. 5a). This arrangement allows the polyprenyl chain of PPM to align along the TM region formed by TM helices 6 and 9. On the opposite side of the active site, the sugar head moiety of the acceptor substrate Ac$_1$PIM4 is positioned near the putative catalytic base D58 with its acyl chains extending towards TM helix 3 (Fig. 5a).

PimE uses PPM as the donor substrate and catalyzes the transfer of a mannosyl moiety from PPM to Ac$_1$PIM4, forming an α(1 → 2) glycosidic bond and leading to the formation of Ac$_1$PIM5. The stereochemical outcome classifies PimE as an inverting glycosyltransferase. Based on our structural observations and the general mechanism of inverting glycosyltransferases[57,58], we propose that D58, located at the

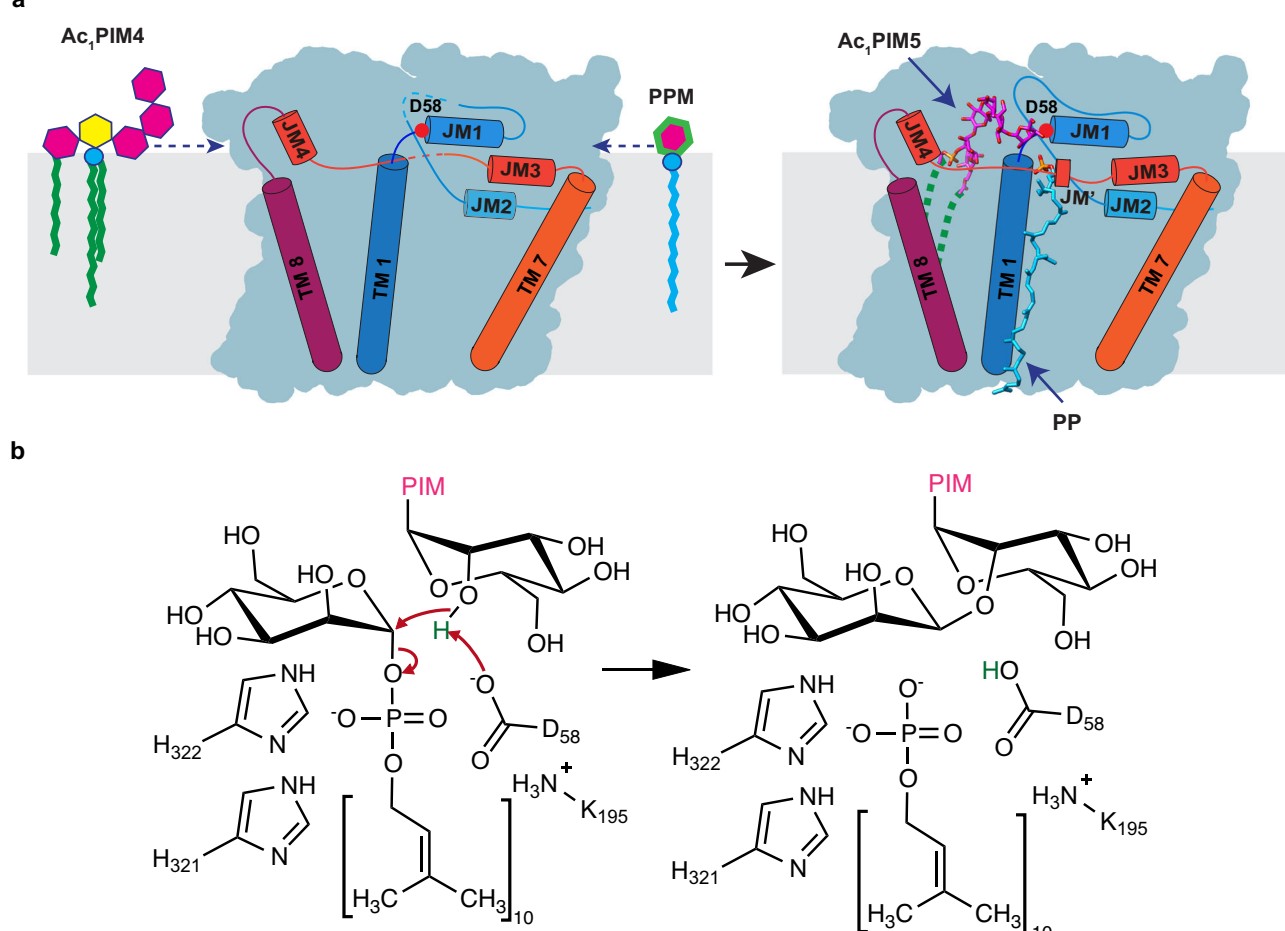

**Fig. 5 | Proposed catalytic mechanism of PimE. a** Schematic representation of the active site of PimE with bound substrates, showing the positioning of PPM and Ac₁PIM4. D58, located at the tip of JM1, acts as the catalytic base. **b** Proposed catalytic mechanism, showing the role of D58 in deprotonating the 2-OH group of Ac₁PIM4, initiating a nucleophilic attack on the anomeric carbon of PPM. This leads to the formation of an α(1 → 2) glycosidic bond between the mannose moiety of PPM and Ac₁PIM4. The phosphate group of PPM is cleaved and stabilized by K195 and H322, forming the product Ac₁PIM5 and the by-product PP. The tetra-acylated acceptor Ac₂PIM4 and its corresponding glycosylated product Ac₂PIM5 are not shown here for the sake of simplicity.

tip of JM1, acts as the catalytic base which deprotonates the 2-OH group of Ac₁PIM4, initiating a nucleophilic attack on the anomeric carbon of PPM (Fig. 5b). This attack leads to the formation of an α(1 → 2) glycosidic bond between the mannose moiety of PPM and Ac₁PIM4. Our structural data and mutational studies suggest that K195 and H322 play a role in stabilizing the negative charge of the phosphate group, facilitating its cleavage from PPM and the formation of Ac1PIM5 and the by-product PP (Fig. 5b).

## Discussion

PIMs are glycolipids present in abundance in the cell envelope of mycobacteria, that play critical roles in bacterial physiology and pathogenesis[18,24,59,60]. The biosynthesis of higher-order PIMs involves a series of mannosylation steps, with PimE catalyzing a late-stage reaction: the addition of the fifth mannose residue to form PIM5[30]. The high-resolution cryo-EM structures of *Ma*PimE in the apo and liganded form presented in this study provide insights into its architecture, ligand-binding properties, and catalytic mechanism. While some peripheral regions of the bound products showed weaker density and should thus be interpreted with caution, the interactions between the bound products and their surrounding protein residues at the center of the active site are clearly resolved.

A distinctive feature of PimE is the presence of a cavity with a shape, and electrostatic and hydrophilic properties that enable it to accommodate both the tetramannoside sugar head of the acceptor substrate Ac₁/₂PIM4 and the donor substrate PPM. Notably, the cavity is flanked by distinct substrate entry points, with PPM accessing the active site through an arch-like structure formed by IL7, placing its long hydrophobic carbon chain along TM helix 6, and Ac₁/₂PIM4 entering from the opposite side, with its sugar head positioned near the catalytic base D58 (Fig. 5a).

PimE belongs to the GT-C_A subclass of glycosyltransferases, characterized by a conserved core module of seven TM helices. Structural comparisons of PimE with other GT-C_A glycosyltransferases, such as ALG6, ArnT, and PglB reveal a conserved mode of binding lipid-linked sugar donors[49,50,61] (Supplementary Fig. 5a–d). In PimE, ALG6, and ArnT, the donor molecules access the active site through an arch-like structure formed by the loop bridging TM helices 7 and 8, while in PglB, this arch is formed by the loop between TM helices 9 and 10. Furthermore, the hydrophobic moiety of the donor substrate interacts with TM helix 6 in all these enzymes.

The GT-C_A architecture of PimE contrasts with the earlier enzymes in PIM biosynthesis, such as PimA and PimB (Fig. 1a). The latter two enzymes are peripheral membrane-associated proteins belonging to the GT-B superfamily, characterized by two Rossmann fold-like domains[26,62]. This structural distinction likely corresponds to the differing nature of their substrates: While PimA and PimB use the soluble guanosine diphosphate (GDP)-mannose[26,27,62,63], PimE uses the lipid-

anchored PPM[52]. It has been speculated that PimE could also be responsible for adding the sixth mannose residue to $Ac_{1/2}PIM5$[30]. We cannot confirm this hypothesis based on our structural and functional data, but it seems plausible, given that both reactions catalyze the formation of an $\alpha(1 \rightarrow 2)$ glycosidic bond to incorporate the mannose residue.

Glycosyltransferases can be broadly classified as either retaining or inverting enzymes based on a comparison of the stereochemistry at the anomeric center in the donor and the newly-formed glycosidic linkage[56,57]. PimE, like many other structurally characterized GT-C enzymes[56], is an inverting glycosyltransferase, which is thought to operate through a single-step nucleophilic substitution mechanism facilitated by an enzymatic general base catalyst and can occur in either a metal-dependent or independent way[57,58]. Our study suggests that PimE operates through a metal-independent mechanism, as supported by our biochemical assays, and by the absence of a DxD motif, typically associated with metal-dependent glycosyltransferases, in the enzyme active site. The metal-independent nature of PimE aligns with observations in structurally related enzymes like ALG6[49] and AftA[52], though it also contrasts with others, such as PglB[51,61]. The metal dependency of earlier enzymes in the PIM biosynthesis pathway, including PimA and PimB, remains unreported. Based on our structural data, combined with computational studies and functional assays, we propose a catalytic mechanism for PimE. We suggest that D58 acts as the catalytic base, deprotonating the hydroxyl group of the acceptor substrate for nucleophilic attack on the anomeric carbon of the sugar donor. In metal-dependent glycosyltransferases, divalent cations often facilitate substrate binding, and stabilize the departing leaving group[58,64]. In the absence of metal coordination, PimE appears to rely on the precise positioning of its catalytic residues and substrates within the active site. The orientation of D58 relative to the substrates is consistent with its proposed role as the catalytic base, while other conserved residues (such as K195 and H322) appear to be involved in substrate orientation and leaving group stabilization. This proposed mechanism provides a starting point for understanding the stereochemical outcome of the reaction catalyzed by PimE.

The role of PimE in the biosynthesis of higher-order PIMs is vital for mycobacterial cell wall integrity and virulence. PimE deletion compromises multiple aspects of mycobacterial fitness, including increased sensitivity to antibiotics and copper[40], altered envelope permeability[39], and defective growth[39,40], highlighting its potential as a target for anti-tuberculosis therapeutics. Our structural characterization of PimE reveals a substrate-binding cavity with its distinct electrostatic properties and conserved residues involved in substrate recognition and catalysis, which could provide a potential binding site for small molecule inhibitors. Moreover, given the high conservation of PIM biosynthetic pathways across mycobacterial species, our findings are likely to extend to other clinically relevant mycobacteria, including *M. tuberculosis*, and could serve as a foundation for structure-based drug design, potentially leading to anti-mycobacterial agents that target this critical cell envelope biosynthetic pathway.

## Methods

### Overexpression and purification of *Ma*PimE
WT *Ma*PimE was cloned into pNYCOMPSC23 plasmid[65]. The procedure for the design and synthesis of pNYCOMPSC23 plasmid was described in detail in a previously published protocol[65]. The plasmids were transformed into BL21 (DE3) pLysS *E. coli* for protein expression. The transformed cells were grown at 37 °C to $OD_{600}$ 0.6–0.8. The culture was then cooled to 22 °C and protein expression was induced by the addition of 0.5 mM IPTG for 16 h. Cells were harvested by centrifugation, the pellet was resuspended in lysis buffer containing 20 mM HEPES pH 7.5, 200 mM NaCl, 20 mM MgSO4, 10 mg/mL DNase I (Roche), 8 mg/mL RNase A (Roche), 1 mM tris(2-carboxyethyl) phosphine hydrochloride (TCEP), 1 mM PMSF, 1 tablet/1.5 L buffer EDTA-free cOmplete protease inhibitor cocktail (Roche). Cells were lysed by passing through a chilled Emulsiflex C3 homogenizer (Avestin) three times. The lysate was centrifuged at 3000×*g* in a Centrifuge 5810 R (Eppendorf) at 4 °C for 5 min to remove cell debris and non-lysed cells. To isolate the cell membrane, the supernatant was ultra-centrifuged in Type 45 Ti Rotor (Beckman Coulter) at 185,600×*g* for 30 min. About 4 L of culture, yielded ~4–6 g of membrane pellet, which were resuspended to a final volume of 80–120 mL with lysis buffer, and homogenized using a handheld glass homogenizer (Kontes) on ice. The membrane fraction was stored at −80 °C until further use. The membrane fraction was thawed and solubilized by adding DDM to a final concentration of 1% (w/v) and was rotated gently at 4 °C for 2 h. Insoluble material was removed by ultracentrifugation at 185,600×*g* in a Type 45 Ti Rotor at 4 °C for 30 min. The supernatant was added to 3 mL pre-equilibrated $Ni^{2+}$-NTA resin (QIAGEN) in the presence of 40 mM imidazole and incubated with gentle rotation at 4 °C for 2 h.

The resin was then washed with 10 column volumes (CV) of wash buffer containing 20 mM HEPES pH 7.5, 200 mM NaCl, 65 mM Imidazole, 0.1% DDM, and eluted with elution buffer containing 20 mM HEPES pH 7.5, 200 mM NaCl, 300 mM Imidazole, 0.03% DDM. The eluted protein was exchanged into a buffer containing 20 mM HEPES pH 7.5, 200 mM NaCl, and 0.03% DDM using a PD-10 desalting column (GE).

### Nanodisc reconstitution of PimE
The protein was then incorporated into lipid nanodisc (Bayburt and Sligar, 2010) with a molar ratio of 1:5:200 of membrane scaffold protein 1E3D1 (MSP1E3D1): 1-palmitoyl-2-oleoyl-glycero-3-phosphocholine (POPC) (Avanti), and incubated for 2 h with gentle agitation at 4 °C. For the substrate-bound structure, $Ac_1PIM4$ and PPM were added to the reconstitution mixture at a molar ratio of 1:10:10 (PimE:$Ac_1PIM4$:PPM). The POPC lipid was prepared by adding the solid extract to deionized water to a final concentration of 10 mM. The mixture was placed on ice and then gently sonicated with a tip sonicator (Fisher Scientific) to dissolve the lipids until it became semitransparent.

To initiate nanodisc reconstitution, 100 mg Biobeads (Bio-Rad) per mL of protein solution were added to the mixture and incubated by gentle rotation at 4 °C overnight. Biobeads were removed the next day by passing the reconstitution mixture through an Ultrafree centrifugal filter unit (Fisher) at 16,100×*g* in a Centrifuge 5415 R (Eppendorf) at 4 °C for 1 min. To remove free nanodisc, the reconstitution mixture was rebound to $Ni^{2+}$-NTA resin with 20 mM imidazole for 2 h at 4 °C. The resin was washed with 10 CV of wash buffer containing 20 mM HEPES pH 7.5, 200 mM NaCl, and 40 mM imidazole, followed by 4 CV of an elution buffer containing 20 mM HEPES pH 7.5, 200 mM NaCl, and 300 mM imidazole. The eluted protein was subsequently purified by size-exclusion chromatography (SEC) with a Superdex 200 Increase 10/300 GL column in buffer containing 20 mM HEPES pH 7.5, 200 mM NaCl.

### Cryo-EM sample preparation
Fractions containing PimE incorporated into nanodiscs were pooled and incubated with Fab-E6 at 4 °C for 2 h at a 1:3 molar ratio of protein to Fab. The protein mixture was further purified by SEC using a Superdex 200 Increase 10/300 GL column run in buffer containing 20 mM HEPES pH 7.5, 200 mM NaCl.

For the apo structure, peak fractions were pooled and concentrated to a final concentration of 5 mg/mL using a 50 kDa filter concentrator (Amicon). The sample was frozen using a Vitrobot (Thermo Fisher) by applying 3 µL of the sample to plasma cleaned (Gatan Solarus) 0.6/1-mm holey gold grid (Quantifoil UltrAuFoil). After a 30-s incubation, the grids were blotted using 595 filter paper (Ted Pella, Inc) for 8 s before being immediately plunged into liquid ethane for vitrification. The plunger operated at 4 °C with greater than 90%

humidity to minimize evaporation and sample degradation. For the substrate-bound structure, peak fractions were pooled and concentrated to a final concentration of 5 mg/mL. The grids were prepared the same way as for the apo sample, but with a blotting time of 9 s.

## Cryo-EM data collection

For the apo structure, images were recorded at the Columbia University Cryo-Electron Microscopy center on a Titan Krios electron microscope (FEI) equipped with an energy filter and a K3 direct electron detection filter camera (Gatan K3-BioQuantum) using a 0.87 Å pixel size. An energy filter slit width of 20 eV was used during the collection and was aligned automatically every hour using the Leginon software package (Suloway et al., 2005). Data collection was performed using a dose of ~58 $e^-$ per $Å^2$ across 50 frames (50 ms per frame) at a dose rate of around 16 $e^-$ per pixel per second, using a set defocus range of $-1.5$ to $-2.6$ μm. A total of 15,198 micrographs were collected over a 2-day session.

For the substrate-bound structure, images were recorded at the New York Structural Biology Center (NYSBC) on an FEI Titan Krios (NYSBC Krios 2) operating at 300 kV equipped with a spherical aberration corrector, an energy filter (Gatan GIF Quantum), and a post-GIF K2 Summit direct electron detector, using a 0.825 Å pixel size. An energy filter slit width of 20 eV was used during the collection and was aligned automatically every hour using the Leginon software package[66]. Data collection was performed using a dose of 50.3 $e^-$ per $Å^2$ across 24 frames (50 ms per frame) at a dose rate of around 41 $e^-$ per pixel per second, using a set defocus range of $-0.8$ to $-2.5$ μm. A total of 21,900 micrographs were collected over a single session of 4 days.

## Identification of MaPimE-specific Fab using phage display

MaPimE was reconstituted into chemically biotinylated-MSP1E3D1 as described above. Selection for Fabs was performed starting with Fab Library E[67,68]. Targets and the library were first diluted in a selection buffer (20 mM HEPES, pH 7.4, 150 mM NaCl, and 1% BSA). Five rounds of sorting were performed using a protocol adapted from published protocols[45,69]. In the first round, bio-panning was performed manually using 400 nM of MaPimE, which was first immobilized onto magnetic beads and washed three times with a selection buffer. The library was incubated for one hour with the immobilized target, beads were subsequently washed three times with selection buffer, and then beads were used to directly infect log phase E. coli XL-1 Blue cells. Phages were amplified overnight in 2XYT media supplemented with ampicillin (100 μg/mL) and M13-K07 helper phage ($10^9$ pfu/mL). To increase the stringency of selection pressure, four additional rounds of sorting were performed by stepwise reduction of the target concentration: 200 nM in the second round, 100 nM in the third round, and 50 nM in the fourth and fifth rounds. These rounds were performed semi-automatically using a KingFisher magnetic beads handler (Thermo Fisher Scientific). For each round, the amplified phage population from each preceding round was used as the input pool. Additionally, amplified phages were pre-cleared prior to each round using 100 μL of streptavidin paramagnetic particles, and 2.0 μM of empty MSP1E3D1 nanodiscs were used throughout the selection as competitors in solution. For rounds 2–5, prior to infection of log-phage cells, bound phage particles were eluted from streptavidin beads by a 15-min incubation with 1% Fos-choline-12 (Anatrace).

## Single-point phage ELISA to validate Fab binding to MaPimE

Ninety-six-well plates (Nunc) were coated with 2 μg/mL Neutravidin and blocked with selection buffer. Colonies of E. coli XL-1 Blue cells harboring phagemids from the fourth and Fifth rounds were used to inoculate 400 μL 2XYT media supplemented with 100 μg/mL ampicillin and $10^9$ pfu/mL M13-KO7 helper phage, and phages were subsequently amplified overnight in 96-well deep-well blocks with shaking at 280 rpm. Amplifications were cleared of cells with a centrifuge step and

then diluted tenfold into ELISA buffer (selection buffer with 2% BSA). All phages were tested against wells with immobilized biotinylated-MSP1E3D1-reconstituted MaPimE (30 nM), empty biotinylated-MSP1E3D1 nanodiscs (50 nM), or buffer alone to determine specific target binding. Phage ELISA was subsequently performed as previously described, where the amount of bound phage was detected by colorimetric assay using an anti-M13 HRP-conjugated monoclonal antibody (GE Healthcare, code 27-9421-01). Binders with high target and low non-specific signals were chosen for subsequent experiments.

## Fab cloning, expression, and purification

Specific binders based on phage ELISA results were sequenced at the University of Chicago Comprehensive Cancer Center DNA Sequencing Facility, and unique clones were then sub-cloned into the Fab expression vector RH2.2 (kind gift of S. Sidhu) using the In-Fusion Cloning kit (Takara). Successful cloning was verified by DNA sequencing. Fabs were then expressed and purified as previously described[70]. Following purification, Fab samples were verified for purity by SDS-PAGE and subsequently dialyzed overnight in 20 mM HEPES, pH 7.4, and 150 mM NaCl.

## Assessment of Fab binding affinity to MaPimE

To measure the apparent binding affinity, multi-point ELISAs using each purified Fab were performed in triplicate. Briefly, MaPimE (30 nM) or empty biotinylated-MSP1E3D1 nanodiscs (50 nM) were immobilized onto 96-well plates coated with Neutravidin (2 μg/mL). Fabs were diluted serially threefold into ELISA buffer using a starting concentration of 3 μM, and each dilution series was tested for binding to wells containing either MaPimE, empty nanodiscs, or no target at all. The Fab ELISA was subsequently performed as previously described, where the amount of bound Fab was measured by a colorimetric assay using an HRP-conjugated anti-Fab monoclonal antibody (Jackson ImmunoResearch). Measured $A_{450}$ values were plotted against the log Fab concentration, and $EC_{50}$ values were determined in GraphPad Prism version 8.4.3 using a variable slope model assuming a sigmoidal dose response.

## Single-particle cryo-EM data processing and map refinement

All data sets were corrected for beam-induced motion with Patch Motion Correction implemented in cryoSPARC v.2.15[71] and the contrast transfer function (CTF) was estimated with Patch CTF. For the apo structure, 5,693,488 particles were automatically picked using a blob-picker job and subjected to multiple rounds of 2D classification. Representative 2D classes of 80,420 particles clearly showing the Fab-bound complex in the side views were selected as input for Topaz training. The resulting model was used to pick particles using Topaz Extract. Initially, 2,137,179 particles were extracted using a 320-pixel size box and binned four times. Multiple rounds of 2D classification were performed to clean up the particles using a batchsize of 400 per class and "Force Max over poses/shifts" turned off with 40 online-EM iterations. Classes with a total of 271,023 particles that displayed clear features of a Fab-bound nanodisc-embedded membrane protein were re-extracted using a 360-pixel box size without binning. Ab initio reconstruction was performed in cryoSPARC v.2.15 using two classes and a class similarity parameter of 0.1. From the ab initio reconstruction, one good class of 174,773 particles was subjected to heterogeneous refinement, and a final class of 145,477 particles was then subjected to non-uniform refinement and yielded a reconstruction with a resolution of 3.24 Å (FSC = 0.143). A subsequent local refinement was performed using a mask covering PimE and the variable region of the Fab and resulted in a density map at 3.02 Å resolution.

For the substrate-bound structure, 12,083,984 particles were automatically picked using a blob-picker job and subjected to multiple rounds of 2D classification. Representative 2D classes of 103,632 particles clearly showing the Fab-bound complex in the side views were

selected as input for Topaz training. The resulting model was used to pick particles using Topaz Extract. 2,196,237 particles were extracted in a box of 320 pixels, and Fourier cropped to 80 pixels for initial cleanup. Multiple rounds of 2D classification were performed to clean up the particles using a batchsize of 400 per class and "Force Max over poses/shifts" turned off with 40 online-EM iterations. Classes with a total of 140,016 particles which displayed clear features of a Fab-bound nanodisc-embedded membrane protein were re-extracted using a 360-pixel box size without binning. Ab initio reconstruction was performed in cryoSPARC v.2.15 using two classes and a class similarity parameter of 0.1. One good class comprises of 78,648 particles were subjected to heterogeneous refinement and a final class comprised of 56,510 particles was subjected to non-uniform refinement and yielded a reconstruction with a resolution of 3.8 Å (FSC = 0.143). A subsequent local refinement was performed using a mask covering PimE and the variable region of the Fab and resulted in a density map at 3.46 Å resolution.

### Model building
Model building was performed in Coot[72]. Most of the Fab, except for the binding interface, was built based on a preexisting model at high resolution (PDB ID 5UCB). The atomic model of apo PimE was built de novo from the globally sharpened 3.02 Å map using the Phenix Map to Model tool. Segments were manually joined, and residues were assigned based on the secondary structure prediction of Xtalpred[73]. Ramachandran outliers were manually fixed in Coot[72] (version 0.9.6 EL). The structure was then refined using Phenix real space refine[74] with secondary structure and Ramachandran restraints imposed. Models were validated using Molprobity within PHENIX 1.19.2[74,75]. For substrate-bound PimE, the model was built based on the apo structure, and the ligand files were generated in Coot (0.9.6 EL) and followed by iterative rounds of real-space refinement in PHENIX 1.19.2 and manual adjustment in Coot (0.9.6 EL).

### Expression and preparation of the PPM synthase
The second domain of the gene encoding a PPM synthase in *Mycobacterium tuberculosis* H37Rv, which starts with a methionine at position 594 and named ppm1_D2[37] was cloned into the expression vector pET23a and introduced in *E. coli* BL21 (DE3) PLysS cells. Cells were grown in a terrific broth medium supplemented with 100 μg/mL ampicillin and 34 μg/mL chloramphenicol at 37 °C with agitation at 180 rpm until the optical density at 600 nm reached 0.5–0.6. At this point, the culture was incubated at 4 °C for 15 min. Protein expression was induced with 0.5 mM IPTG, and growth was allowed to proceed at 16 °C for 16 h. Cells were harvested by centrifugation at 2500×*g* for 10 min at 4 °C and resuspended in 25 mM Tris/HCl pH 7.6 supplemented with 10 mM MgCl₂, and 5 mM β-mercaptoethanol. Cells were disrupted by sonication with cycles of 30 s of ice/disruption, repeated four times. Cell debris was removed by centrifugation at 3000×*g* for 30 min at 4 °C, and the supernatant was ultracentrifuged at 84,000×*g* for 1 h at 4 °C. The supernatant was discarded, and the membrane fraction was resuspended gently in 25 mM Tris/HCl pH 7.6 supplemented with 10 mM MgCl₂ and 5 mM β-mercaptoethanol. This membrane fraction was further used to produce PPM.

### Production of non-radiolabeled PPM
The membrane fraction of *E. coli* cells, harboring the PPM synthase gene, was incubated with 2.5 mM GDP-mannose, 0.1 mM ATP, 4 mM MgCl₂, 50 mM Bicine buffer pH 8.0, and 1 mg of decaprenyl monophosphate (Larodan AB, Sweden) solubilized in CHAPS. The final concentration of CHAPS was 1% and the reaction mixture was at a final volume of 2.5 mL. The mixture was incubated for 1 h at 37 °C and the reaction quenched with the addition of 20.5 mL CHCl₃/CH₃OH/0.8 M NaOH (10:10:3, v/v/v), and incubation at 50 °C for 20 min. The mixture was allowed to cool down and purification of PPMs was carried out.

Briefly, 1.75 mL of CHCl₃ and 0.75 mL of H₂O were added, and the mixture vortexed. The upper phase (aqueous phase) was discarded, and the bottom phase was washed three times with 2 mL of CHCl₃/CH₃OH/H₂O (3:47:48, v/v/v). The lower phase was dried under a nitrogen stream and the residue was resuspended in CHCl₃/CH₃OH/H₂O (90:70:20, v/v/v). The PPM solution was loaded onto a silica gel 60 column (2.5 cm × 20 cm), prepared with 20 g of silica gel equilibrated with the same solvent system. Elution was carried out with the same solvent mixture: twenty fractions of 2 mL each were collected. The fractions were analyzed by High-Performance Thin-Layer chromatography (HPTLC) by applying 10 μL of each. Visualization was performed, first by spraying with primulin and then with orcinol[54]. The fractions containing pure PPM were pooled and dried under a nitrogen stream. The residue was solubilized in 200 μL of CHCl₃/CH₃OH (2:1). The amount of PPM was estimated by total phosphorus quantification[76].

### Production of radiolabeled PPM
The production of PP[U-¹⁴C]M was carried out as described for non-radiolabeled PPM, with the following modifications. The volume of the reaction mixture was decreased to 350 μL; the amount of decaprenyl monophosphate was 0.14 mg in 5% CHAPS. The final concentration of CHAPS in the reaction was 1%. Also, GDP-[U-¹⁴C] mannose (300 mCi/mmol) was added at a final concentration of 7.6 μM. The enzyme inactivation and purification of labeled PPM was carried out as described above, but the chromatographic step was excluded. The resulting PPM was resuspended in 200 μL of CHCl₃/CH₃OH (2:1, v/v), and quantification was done by scintillation counting. The absence of GDP-[U-¹⁴C] mannose was confirmed by TLC and the subsequent analysis was done by using the phosphor imaging plate and Fuji FLA-5100 imaging system.

### Purification of Ac₁PIM4
Ac₁PIM4 was purified from *M. smegmatis* mc²155 ΔpimE as previously described[54]. Briefly, cells were grown in Luria-Bertani (LB) medium supplemented with kanamycin (40 μg/mL) at 37 °C and harvested, at stationary phase, by centrifugation. Total lipids were extracted with chloroform/methanol (1:1, v/v). Following centrifugation, the chloroform layer, containing the desired membrane lipids, was collected and dried. The residue was subjected to biphasic partitioning between chloroform and water to isolate phospholipids. The chloroform phase was dried and treated with hot acetone to precipitate PIMs. The precipitate was subjected to sequential chromatographic purification steps. First, a Silica-Gel 60 column was used and PIMs (Ac₁PIM₄ and Ac₁PIM₂) were eluted with CHCl₃/CH₃OH (1:1, v/v). Then, HPLC was performed with a C18 column and Ac₁PIM₄ was eluted with a gradient of the following solvents (CHCl₃/CH₃OH (7:3, v/v) with 50 mM NH₄CH₃COO, and CHCl₃/CH₃OH/H₂O (24:114:62, v/v/v) with 10 mM NH₄CH₃COO). Finally, a second Silica-Gel 60 column was used, and pure Ac₁PIM₄ was eluted with CHCl₃/CH₃OH/25% NH₃/1 M NH₄CH₃COO in water. Throughout the purification process, compounds were identified by thin-layer chromatography (TLC) using chloroform/methanol/25% NH₃/1 M NH₄CH₃COO in water/H₂O (180:140:9:9:23, v/v/v/v/v) as the mobile phase. The spots on TLC plates were visualized by spaying with primuline (for lipids) and orcinol (for carbohydrates). The purified Ac₁PIM₄ was characterized by ESI-MS in negative mode, showing the expected [M-H]⁻ ion at m/z 1737.9, and its structure was confirmed by ¹H-NMR spectroscopy in CDCl₃/CD₃OD/D₂O (70:30:2, v/v/v). The product was quantified using a total phosphorus quantification protocol adapted from a previously described method[76].

### Determination of PimE activity in the membrane fraction
*E. coli* BL21 (DE3) pLysS cells harboring the *Mycobacterium abscessus* PimE gene (*Ma*PimE) were grown as described above. The cell pellet

was resuspended in 25 mM HEPES pH 7.6 and disrupted by sonication using a tip sonicator (4 cycles of 30 s pulses with intervals of 30 s between pulses). Cell debris was removed by centrifugation (3000×$g$, 30 min, 4 °C). The supernatant was ultracentrifuged (84,000×$g$, 1 h, 4 °C), and the membrane fraction was gently resuspended in 25 mM HEPES pH 7.6. Total protein content was determined with the Pierce BCA protein assay kit. The $K_M$ and apparent $V_{max}$ of $Ma$PimE for PPM were evaluated in reaction mixtures (final volume, 30 μL), containing 50 mM Bicine buffer pH 8.0, 1 mM MgCl$_2$, 1 mM Ac$_1$PIM4, and different concentrations of PPM (50 μM to 1 mM; 2 μM PP[U-$^{14}$C]M was included for each concentration examined). The $K_M$ and apparent $V_{max}$ for Ac$_1$PIM4 was assessed in reaction mixtures (final volume, 3 μL), containing 50 mM of Bicine buffer pH 8.0, 1 mM MgCl$_2$, 0.998 mM PPM, 2 μM PP[U-$^{14}$C]M, and different concentrations of Ac$_1$PIM4 (50 μM to 1 mM). Both substrates (PPM and Ac$_1$PIM4) were solubilized in DDM, 1% (w/v) final concentration. The reaction mixtures were preincubated at 37 °C for 1 min, and the reaction was initiated by the addition of the membrane fraction (150 μg total protein) and incubated for a further 6 min; reactions were stopped by heating at 80 °C for 10 min. It was confirmed that the product formation is linear for up to 9 min of reaction time; furthermore, no Ac$_1$PIM5 was produced during the enzyme inactivation step (Supplementary Fig. 9a, b).

Aliquots of the reaction mixtures (20 μL) were loaded onto an HPTLC plate and developed with chloroform/methanol/13 M ammonia/1 M ammonium acetate/water (180:140:9:9:23, v/v/v/v/v). The production of Ac$_1$PIM5 was quantified by exposing, simultaneously, an HPTLC plate containing a known concentration of [U-$^{14}$C]glucose (7.81 × 10$^{-4}$ μCi). The HPTLC plate was exposed for 24 h to a phosphor imaging plate; the image was obtained with a FujiFilm FLA-5100 imaging system. Quantification of the radiolabeled product, Ac$_1$PIM5, was performed using the Fiji software[77]. $K_M$ and apparent $V_{max}$ values and respective standard deviations were calculated using the Origin software for nonlinear regression according to the Michaelis–Menten equation. All the reactions were performed in duplicate[54].

### In vitro test of activity of $Ma$PimE mutants

Point mutants of $Ma$PimE were generated using the QuikChange site-directed mutagenesis kit (Agilent) (Supplementary Table 2). $E.\ coli$ BL21 (DE3) cells harboring the WT and mutated genes were grown, and membrane fractions were prepared as described above. Western blot analysis showed that all mutants have expression levels similar to the WT $Ma$PimE (Supplementary Fig. 12d). The activity of mutants and WT was evaluated under substrate saturation conditions (1 mM of each substrate) as described for the kinetic characterization of $Ma$PimE.

### Metal-dependency assay for $Ma$PimE activity

$E.\ coli$ membrane fractions expressing $Ma$PimE were prepared as described above. Both substrates (PPM and Ac$_1$PIM4) were solubilized in 1% (w/v, final) DDM. Reaction mixtures (30 μL) contained 50 mM Bicine buffer (pH 8.0), membrane fraction (150 μg total protein), and substrates PPM and Ac$_1$PIM4 (prepared as described below). To assess metal requirements, reactions were supplemented with either metal chelators (5 mM each of EGTA and EDTA) or 10 mM MgCl$_2$. Control reactions were performed without chelators or added metals. Reactions were incubated at 37 °C for 4 h, then stopped by heating at 95 °C for 5 min. Samples (3 μL) were spotted onto silica gel 60 HPTLC plates and developed using chloroform/methanol/13 M ammonia/1 M ammonium acetate/water (180:140:9:9:23, v/v/v/v/v) as the mobile phase. Glycolipids were visualized by orcinol staining.

### Mycobacterial strains and growth conditions

WT $M.\ smegmatis$ mc$^2$155[78] and the Δ$pimE$ mutant strain[30] were used in this study. The Δ$pimE$ strain was complemented with pYAMaPimE (encoding WT $M.\ abscessus$ PimE) or its mutant derivatives. Strains were grown in Middlebrook 7H9 broth (Difco) supplemented with 10%

DC (0.85% NaCl, 2% glucose), 0.05% Tween-80, and appropriate antibiotics (20 μg/mL kanamycin for Δ$pimE$; 20 μg/mL kanamycin plus 20 μg/mL streptomycin for complemented Δ$pimE$ strains). Cultures were incubated at 37 °C with shaking at 180 rpm until mid-log phase (OD$_{600}$ 0.5–1.0).

### Construction of complementation plasmids and site-directed mutagenesis

The $M.\ abscessus$ $pimE$ gene was amplified by PCR and cloned into pYAB186[79], replacing the $M.\ smegmatis$ $pimE$ gene to create the new plasmid pYAMaPimE. The pYAB186 vector contains both Flag and GFP tags[79], which were retained in pYAMaPimE, resulting in Flag-GFP-tagged $Ma$PimE. Site-directed mutagenesis (Supplementary Table 2) was performed using the QuikChange Lightning Kit (Agilent Technologies) following the manufacturer's instructions. All constructs were verified by DNA sequencing. The plasmid pYAB186 was kindly provided by Dr. Yasu S. Morita from the University of Massachusetts.

### Transformation of $M.\ smegmatis$

Electrocompetent $M.\ smegmatis$ Δ$pimE$ cells were prepared and transformed as described previously[78]. Transformants were selected on Middlebrook 7H10 agar supplemented with 10% DC and appropriate antibiotics.

### Lipid extraction and HPTLC analysis

Lipid extraction and analysis were performed following an established protocol[80]. Briefly, bacterial cultures (50 OD$_{600}$ units) were harvested by centrifugation at 3214×$g$ for 10 min. Cell pellets were subjected to a three-step extraction process: first with 20 volumes of chloroform/methanol (2:1, v/v) for 1.5 h, followed by two extractions with 10 volumes each of chloroform/methanol (2:1, v/v) and chloroform/methanol/water (1:2:0.8, v/v/v) for 1.5 h each at room temperature. The combined organic extracts were dried under an N$_2$ stream and resuspended in water. The butanol phase was dried and resuspended in water-saturated butanol, and 3 μL (equivalent to 3 mg cell dry weight) were spotted on HPTLC silica gel 60 plates (Merck) and developed in chloroform/methanol/13 M NH$_3$/1 M NH$_4$Ac/water (180:140:9:9:23, v/v/v/v/v) for ~2 h. Plates were dried and sprayed with orcinol spray reagent (0.1% orcinol in 20% H$_2$SO$_4$) and heated to visualize glycolipids.

### Western blot analysis

Wild-type, Δ$pimE$, and complemented $M.\ smegmatis$ were cultured as described above. Log phase cells (OD$_{600}$ = 0.5–1.0) were harvested by centrifugation and washed once with 50 mM HEPES buffer (pH 7.4). Cell pellets were resuspended in lysis buffer (25 mM HEPES pH 7.4, 2 mM EGTA) supplemented with an EDTA-free cOmplete protease inhibitor cocktail (Roche). Cells were lysed by five passages through a chilled Emulsiflex C3 homogenizer (Avestin). The lysate was centrifuged at 10,000×$g$ for 15 min at 4 °C to remove cell debris and unbroken cells. The supernatant was then ultracentrifuged in a Type 70 Ti Rotor (Beckman Coulter) at 126,000×$g$ for 30 min at 4 °C to isolate the membrane fraction. The pellet was resuspended in lysis buffer, and total protein content was determined using the Pierce BCA protein assay kit (Thermo Fisher Scientific). Equal amounts of protein (20 ng per lane) were loaded for SDS-PAGE, and proteins were transferred to PVDF membranes using the Trans-Blot Turbo Transfer System (Bio-Rad) following the manufacturer's instructions. Membranes were briefly stained with 0.1% (w/v) Ponceau S in 5% acetic acid to assess equal loading and transfer quality, then destained with distilled water before immunodetection. Flag-GFP-tagged PimE was detected using a mouse monoclonal ANTI-FLAG M2-Peroxidase (HRP) antibody (Sigma-Aldrich, catalog number A8592) at a 1:5000 dilution. Immunoreactive bands were visualized using the ECL Prime Western Blotting Detection Reagent (GE Healthcare) and imaged using Azure 600 ultimate Western blot imaging system (Azure Biosystems).

The membrane fractions of the wild-type *Ma*PimE and *Ma*PimE mutants expressed in *E. coli* were prepared as described above, and total protein content was determined using the Pierce BCA protein assay kit (Thermo Fisher Scientific). After electrophoresis on SDS-PAGE, proteins were transferred onto a PVDF membrane using the Trans-Blot Turbo Transfer System (Bio-Rad). His-tagged PimE mutants were directly detected with HisProbe-Horseradish Peroxidase (HRP) conjugate (1:5000 dilution; Thermo Fisher Scientific). His-tagged proteins were visualized using the ECL Prime Western Blotting Detection Reagent (GE Healthcare) in a ChemiDoc MP Imaging System (Bio-Rad).

**Coarse-grained MD simulations**

The CG parameters for PPM and Ac$_1$PIM4 were generated based on previously published Martini 3 parameters[81–84]. For CG simulations, the protein was converted to the Martini 3 forcefield using martinize2[85], including a 1000 kJ mol$^{-1}$ nm$^{-2}$ elastic network with default protonation states. PimE was then embedded into a PE:PG (80:20) membrane using the insane[86] program, which was followed by solvation with martini water and neutralized with 150 M NaCl. For each of the substrate simulations, 4% of the upper leaflet of the membrane was this lipid, with the placement differing for each simulation. Minimization of the system was achieved using the steepest descents method. A timestep of 20 fs was used for production simulations, with five independent simulations of 5 μs for each substrate. In each simulation, the system was comprised of ~25,000 particles (~15,000 of these being water molecules) in a 14 × 14 × 14 nm$^3$ box. The C-rescale barostat[87] was set at 1 bar, and the velocity-rescaling thermostat[88] was used at 310 K. The Reaction-Field algorithm was used for electrostatic interactions with a cut-off of 1.1 nm. A single cut-off of 1.1 nm was used for van der Waals interactions. All simulations were performed using Gromacs 2021.4[89,90]. For analysis, PLUMED[91] was used in conjunction with matplotlib[92] to generate the density plots. All of the production simulations were used for analysis.

**Atomistic simulations**

Atomistic coordinates were generated from the end snapshots of the CG simulations using CG2AT2[93]. Molecular coordinates of both substrates and products were built using the coordinates from the ligand-bound cryo-EM structure. For the substrate state, the transferred mannose was covalently added to the coordinates of the polyprenyl product, and energy was minimized to form coordinates for the PPM. RoseTTAFold All-atom was used to fold and dock both substrates and products for comparison. Atomistic molecular simulations were performed with the CHARMM36m forcefield, with parameters for Ac$_1$PIM4 and Ac$_1$PIM5 taken from CHARMM-GUI Membrane Builder[82,94,95]. Parameters for the PP and PPM were created using CHARMM-GUI and combined with previous parameters from our past studies. Energy minimizations of the systems were performed using the steepest descents method. In each simulation, the system was comprised of ~135,000 particles (~30,000 of these being water molecules) in an 11 × 11 × 11 nm$^3$ box.

A timestep of 2 fs was used for production simulations, with three independent simulations of 500 ns for substrate, product, and apo states of PimE. C-rescale barostat[87] was set at 1 bar, and the velocity-rescaling thermostat[88] was used at 310 K. The PME algorithm was used for electrostatic interactions with a cut-off of 1.2 nm. A single cut-off of 1.2 nm was used for van der Waals interactions. All simulations were performed using Gromacs 2021.4[89,90].

For the analysis, all the simulations were used. Analysis for the ligand contacts with the protein were performed with MDAnalysis[96]. The RMSF of the protein and RMSD of the ligands were measured using gmx tools. The $pK_a$ of the residues were measured with PROPKA3[97] and propkatraj (version 1.1.0). Plots were created with maplotlib[92] and molecular visualization was performed with PyMOL[98].

**Reporting summary**

Further information on research design is available in the Nature Portfolio Reporting Summary linked to this article.

## Data availability

The cryo-EM maps and atomic coordinates have been deposited with accession numbers EMD-46999 and PDB 9DM7 for apo PimE, and EMD-46998 and PDB 9MJB/9DM5 for product-bound PimE (with Fab_E6 modeled and unmodeled, respectively). The starting and final coordinates and parameters for the MD simulations are available at https://doi.org/10.5281/zenodo.14916601. PDB codes of previously published structures used in this study are 6SNH, 5F15, 5OGL, 7WLD, and 5UCB. Source data are provided as a Source Data file. Source data are provided with this paper.

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

## Acknowledgements

This work was supported by R35GM132120 (to F.M.). P.J.S. acknowledges the NIH (R01AI174416 (PI: M. Stephen Trent)), Wellcome (208361/Z/17/Z), MRC, BBSRC, and the Howard Dalton Centre for funding. C.M.B. was supported by an MRC studentship (MR/N014294/1). P.J.S. and C.M.B. acknowledge Sulis at HPC Midlands+, which was funded by the EPSRC on grant EP/T022108/1, and the University of Warwick Scientific Computing Research Technology Platform for computational access. YSM acknowledges the NIH (R21AI168791) for funding. A.A.K. acknowledges support from NIH grant R01GM117372. This project made use of time on ARCHER2 and JADE2 granted via the UK High-End Computing Consortium for Biomolecular Simulation, HECBioSim (http://www.hecbiosim.ac.uk), supported by EPSRC (grant no. EP/R029407/1). H.S.

acknowledges Fundação para a Ciência e a Tecnologia (PTDC/BIA-BQM/31031/2017 (Lisboa-01-0145-FEDER-031031), MOSTMICRO-ITQB, UIDB/04612/2020 and UIDP/04612/2020). T.L.L. thanks Academia Sinica for support. We thank Sara Rebelo for major technical assistance.

## Author contributions

Y.L. carried out protein expression, purification, cryo-EM sample preparation, data processing, model building, and structural refinement. B.K. conducted the high-throughput screening. M.B.D. and S.G. performed initial detergent screening to optimize purification conditions for *Ma*PimE. Y.L. performed site-directed mutagenesis for *Ma*PimE expressed in *E. coli*. R.N.N., N.B., A.M.E., and C.G.T. designed strategies and performed substrate production and purification as well as in vitro functional assays under the supervision of H.S. C.G.T. constructed the strain for PPM production and optimized growth conditions for PIMs production. Y.L. conducted HPTLC assays for metal-dependency testing of *Ma*PimE under the guidance of N.B. Y.L. performed molecular cloning for all *Ma*PimE constructs expressed in *M. smegmatis* and functional assays, including mycobacterial complementation tests and HPTLC-based analysis of PIMs. Y.S.M. provided WT *M. smegmatis* mc²155, Δ*pimE* mutant strain, pYAB186 plasmid, and the competent cells for mycobacterial complementation tests, along with critical guidance for these experiments. C.M.B. and P.J.S. performed all molecular dynamics simulations and docking studies. S.E. and P.T. identified, characterized, and purified the Fabs under the supervision of A.A.K. T.L.L. provided input and insightful discussions that contributed to the improvement of this manuscript. Y.L. drafted the manuscript with critical contributions from R.N., F.M., P.J.S., C.M.B., T.L.L., Y.S.M., N.B., A.M.E., H.S., S.E., and R.J.C. F.M., R.N., P.J.S., and H.S. supervised the project.

## Competing interests

The authors declare no competing interests.

## Additional information

¹Department of Physiology and Cellular Biophysics, Columbia University Irving Medical Center, New York, NY, USA. ²School of Life Sciences and Department of Chemistry, University of Warwick, Coventry, UK. ³Groningen Biomolecular Sciences and Biotechnology Institute and Zernike Institute for Advanced Materials, University of Groningen, Nijenborgh, The Netherlands. ⁴Instituto de Tecnologia Química e Biológica António Xavier, ITQB NOVA, Universidade Nova de Lisboa, Oeiras, Portugal. ⁵Marine and Environmental Sciences Centre, Escola Superior de Tecnologia, Instituto Politécnico de Setúbal, Setúbal, Portugal. ⁶Department of Biochemistry and Molecular Biophysics, University of Chicago, Chicago, IL, USA. ⁷Department of Chemistry, University of Virginia, Charlottesville, VA, USA. ⁸School of Medicine, New York University, New York, NY, USA. ⁹Institute for Molecular Bioscience, The University of Queensland, St. Lucia, QLD, Australia. ¹⁰Institute of Biological Chemistry, Academia Sinica, Nangang, Taipei, Taiwan. ¹¹Department of Chemistry, University of Alberta, Edmonton, AB, Canada. ¹²Institute of Biochemical Sciences, National Taiwan University, Taipei, Taiwan. ¹³Department of Microbiology, University of Massachusetts, Amherst, MA, USA. ¹⁴Department of Radiation Oncology, Weill Cornell Medicine, New York, NY, USA. ✉e-mail: santos@itqb.unl.pt; Phillip.Stansfeld@warwick.ac.uk; rin7007@med.cornell.edu; fm123@cumc.columbia.edu

