## [Transparent Peer Review file · Nature Communications]

Mechanistic studies of mycobacterial glycolipid biosynthesis by the mannosyltransferase PimE

Corresponding Author: Professor Filippo Mancia

Version 0:

Reviewer comments:

Reviewer #1

(Remarks to the Author)

The manuscript by Liu et al. describes the purification, structural, and biochemical characterisation of PimE. The authors should be congratulated on an excellent study that significantly impacts the field. While overall, the data appear to support the manuscript's conclusions, I have a few major issues that I think should be addressed and a small number of minor suggestions.

The authors have made the puzzling decision to bury all of the careful biochemical work in the supplementary materials of the manuscript. While I recognise the desire to showcase the structural work, it only has significant meaning because the biochemical assays and mutagenesis work back it up. Otherwise, the amount we learn about PimE would be substantially less. I would encourage the authors to include a new figure or extend an existing figure to include the kinetic and TLC-based assays (or a subset of the most important ones) in the main text.

None of the western blots appear to have loading controls or Coomassie gels. The authors conclude that the blots indicate equal expression, but without these controls, it is impossible to infer from the data presented.

The authors state that the enzyme is not metal-dependent. However, there are many examples in the literature where EDTA fails to inhibit metalloenzymes because the protein affinity for the metal is too high. Other, more direct methods exist to measure this, but I don't see this as necessary. While I agree that PimE is unlikely to be metal-dependent, especially in light of the Bloch et al. paper on Alg6, this statement needs to be tempered slightly. The authors may also consider commenting whether, like PimE also lacks a DxD motif generally associated with GT binding of metals as was also observed for Alg6 to further strengthen their conclusion here.

Finally, the fit of the ligand in the density for the mannose residue and phosphate furthest from the reaction centre is very poor. Perhaps I am wrong, but I am not convinced these have been accurately modelled here?

Minor:

Line 30 'its' wrong word

Line 51 is redundant. If it is ancient, it has necessarily 'plagued humanity throughout the ages'

Line 267 The computational data supports this conclusion but does not confirm it. The software helps generate hypotheses, which can be tested using real, direct measurements. Otherwise, there would be no point in structural biology; we could all just rely on AlphaFold.

As discussed above, Figures 3 and 4 should be combined into one more condensed figure and some of the biochemical analysis brought into the main text. Biochemistry effectively holds together most of the work and underpins all of the hypotheses derived from the structural data. It should be in the main section of the paper.

404 enzymes (not enzyme)

The method of purification and confirmation of AcPIM4 is referred to as already published. Given that this is such a core technique to the paper, this should be described in the manuscript methods.

Reviewer #2

(Remarks to the Author)

In this work, Liu et al have determined PimE (a mannosyltransferase catalyzing Ac1/2PIM4 and PPP to Ac1/2PIM5 and PP in the periplasmic side of the mycobacterial inner membrane) cryo-EM structures in apo and the product (Ac1/2PIM5 and PP) bound forms, and utilized modeling and molecular dynamics (MD) simulations, as well as extensive functional studies to better understand the PimE mechanism. Compared to NCOMMS-24-53156-T (another manuscript of Liu et al), this manuscript was written much carefully with convincing ample functional studies.

Line 145: There is no Fab in Extended Data Fig. 3a. Should it be Extended Data Fig. 1g?

Line 176: It is very difficult to read "yellow" texts in Extended Data Fig. 4.

Line 187: Fig. 2d-e -> Fig. 2d-e and Extended Data Fig. 5; In the caption of Fig. 2, (e) and (f) need to be swapped.

Figure 4: For both 4e and f, would it be better to have the same orientation as in Fig. 2c and add "CE1" and "CE2" in the figure?

The RoseTTAFold models: It is not clear described how the authors obtained these models. Which templates were used? It is really surprising to get the similar key interactions between PimE and products, compared to the cryo-EM structure. What additional information the authors got from these models?

Extended Data Fig. 8 and Line 294: 1. Can a ligand RMSD of 0.6 nm be considered stable? 2. How a ligand RMSD can be indicative of ligand affinity?

Extended Data Fig. 9 and Line 309: The RMSF plot should be twice wider in X. More importantly, how the calculated pKa values of D58 and K195 can suggest a possible role in catalysis?

Reviewer #3

(Remarks to the Author)

In this manuscript, Liu et al. present single particle cryo-EM structures of the mycobacterial mannosyltransferase PimE in apo and product-bound states. The authors also thoroughly investigate residues involved in catalysis and substrate/product binding using in vitro and in vivo functional assays and molecular dynamics simulations. Overall, the study is well-executed, well-presented and provides important mechanistic insights into the function of PimE and mycobacterial cell envelope biogenesis that will be of interest to a broad scientific audience. I only have minor issues listed below.

Comments regarding the structures:

1. The details for cryo-EM data collection described in the methods and shown in Table 1 are not consistent. Please check.
2. The particle distribution plots in ED Figure 2 indicate potential preferred orientation problems. Please provide 3DFSC curves.
3. The authors do not present a structural comparison between the apo and product bound structures. What is the C α RMSD, and does this say anything about the mechanism?
4. Why was the Fab not modelled in the product bound structure?
5. Have the authors tried using classification with a mask focusing on the substrate/product cavity to discard any potential particle populations without bound products? This might improve the density of the products.
6. The reaction products are "stuck" on PimE in the structure—how do the authors envisage that the products are released from PimE in vivo, i.e. how is enzyme inhibition by the products prevented?
7. The authors frame the article by highlighting that PimE could be an antibiotic target against *M. tuberculosis*. Can the authors provide a comparison of their structures with a predicted model of *M. tuberculosis* PimE? Are all important residues identified in this study present in the *M. tuberculosis* homologue?
8. Can the authors provide the Z-scores for the structural homologues identified in the DALI search?

Other comments:

9. Have the authors checked if the Fab used for facilitating structure determination has any effect on the catalytic activity? This might tell something about the catalytic mechanism, e.g. if any conformational changes are required.
10. The metal-dependence assay is not comprehensive and relies on chelators to strip PimE of any metals acquired from the expression host rather than adding individual metals and looking for an increase in activity. PimE is clearly still active in the presence of chelators, as shown in ED Figure 1b, but the assay doesn't tell anything about the reaction rate, which might be stimulated by a metal ion. Considering available structural information on metal-dependent glycosyltransferases, is there any putative metal binding site(s) near D58?
11. The authors suggest that PimE is a potential antibiotic target, but a previous study (PMID: 16803893) showed that a pimE deletion mutant is still viable and has an intact cell wall. Is there any additional evidence in the literature that PimE is a promising antibiotic target?
12. The authors highlight the structural similarity of PimE and the human GPI transamidase complex, but also suggest that PimE is a good antibiotic target. These statements seem to be at odds—given that both enzymes catalyse reactions involving phosphatidylinositol mannosides, any PimE-targeting antibiotic might also inhibit the human enzyme. Can the

authors also clarify how PimE is different to the GPI transaminase?

13. L937 – please use either the single- or three-letter amino acid code consistently.

Version 1:

Reviewer comments:

Reviewer #1

(Remarks to the Author)

The authors have addressed all of my concerns. I want to congratulate them on a very nice paper.

Reviewer #2

(Remarks to the Author)

The authors have addressed all my comments and questions properly.

Reviewer #3

(Remarks to the Author)

Thanks to the authors for positively addressing my and other reviewers' comments. The clarity of the manuscript is now much improved and I have no further issues.

Response to Reviewers

Liu *et al.* “Mechanistic studies of mycobacterial glycolipid biosynthesis by the mannosyltransferase PimE”.

We are grateful for the overall positive and constructive feedback from the Reviewers, and for their appreciation of our work. Here we provide a point-by-point response to their comments.

Reviewer #1:

The manuscript by Liu et al. describes the purification, structural, and biochemical characterisation of PimE. The authors should be congratulated on an excellent study that significantly impacts the field. While overall, the data appear to support the manuscript's conclusions, I have a few major issues that I think should be addressed and a small number of minor suggestions.

We are grateful to the Reviewer for their appreciation of our work.

The authors have made the puzzling decision to bury all of the careful biochemical work in the supplementary materials of the manuscript. While I recognise the desire to showcase the structural work, it only has significant meaning because the biochemical assays and mutagenesis work back it up. Otherwise, the amount we learn about PimE would be substantially less. I would encourage the authors to include a new figure or extend an existing figure to include the kinetic and TLC-based assays (or a subset of the most important ones) in the main text.

We appreciate this comment, and in response have included key biochemical results in a main figure. Specifically, we have added the lipid profile analysis by TLC for the *in vivo* mycobacterial complementation assays as Fig. 4g-i.

None of the western blots appear to have loading controls or Coomassie gels. The authors conclude that the blots indicate equal expression, but without these controls, it is impossible to infer from the data presented.

We have updated Extended Data Fig. 10 to include Ponceau S-stained membranes alongside the Western blot images. The Ponceau S staining confirms – within margins of error, consistent with the not exactly quantitative nature of the technique – equal gel loading and high-quality membrane transfer for all samples, supporting the comparability of expression levels between wild-type and mutant *MaPimE*. These updates provide additional confidence in our interpretation of the Western blot data. The revised figure and accompanying legend reflect these changes (lines 952-956), and the Methods section (lines 795-800) has been updated to include a description of the staining procedure.

The authors state that the enzyme is not metal-dependent. However, there are many examples in the literature where EDTA fails to inhibit metalloenzymes because the protein affinity for the metal is too high. Other, more direct methods exist to measure this, but I don't see this as necessary. While I agree that PimE is unlikely to be metal-dependent, especially in light of the Bloch et al. paper on Alg6, this statement needs to be tempered slightly. The authors may also consider commenting whether, like PimE also lacks a DxD motif generally associated with GT binding of metals as was also observed for Alg6 to further strengthen their conclusion here.

To address this relevant and interesting comment, we performed an additional experiment with Mg^{2+} , a common glycosyltransferase cofactor, which showed no apparent effect on product

formation (Extended Data Fig. 1b, right). While this observation may also support that PimE will unlikely require metal ions for activity, we have tempered our statement regarding PimE's metal dependency and included a discussion on the absence of the DxD motif in the enzyme's sequence (lines 374-376). These new experimental findings and their interpretation have been incorporated into the revised Results (370-376) and Discussion (line 435-439) sections.

Finally, the fit of the ligand in the density for the mannose residue and phosphate furthest from the reaction centre is very poor. Perhaps I am wrong, but I am not convinced these have been accurately modelled here?

We acknowledge that the density for the mannose residue and phosphate group furthest from the reaction center is indeed weaker compared to the core region, likely due to increased flexibility or partial occupancy in these distal regions. The model we present here represents the best fit achievable given the current data. We have updated the manuscript to acknowledge this limitation in both the Results (line 246-248) and Discussion (line 402-405) sections. Importantly, we would like to emphasize that the key interactions and mechanistic insights we derive from the structure are based on the well-resolved portions of the ligand and protein. The less well-defined regions do not significantly impact our main conclusions.

Minor:

Line 30 'its' wrong word

We have corrected this (line 31).

Line 51 is redundant. If it is ancient, it has necessarily 'plagued humanity throughout the ages'

We have revised this sentence to remove the redundancy (lines 54-55).

Line 267 The computational data supports this conclusion but does not confirm it. The software helps generate hypotheses, which can be tested using real, direct measurements. Otherwise, there would be no point in structural biology; we could all just rely on Alphafold.

We agree with this point and have modified the wording to more accurately reflect the impact of computational data on our work (line 294-298).

As discussed above, Figures 3 and 4 should be combined into one more condensed figure and some of the biochemical analysis brought into the main text. Biochemistry effectively holds together most of the work and underpins all of the hypotheses derived from the structural data. It should be in the main section of the paper.

We appreciate this thoughtful suggestion. While we understand the value of connecting structural and biochemical data, combining Fig. 3 and 4 would result in an overly dense presentation that might compromise clarity, as each figure contains multiple detailed panels. Instead, as discussed above, we have incorporated key functional assays into Fig. 4. We think this revised organization maintains visual clarity, while properly highlighting biochemical evidence that supports our structural findings.

404 enzymes (not enzyme)

This has been corrected (now in line 432).

The method of purification and confirmation of AcPIM4 is referred to as already published. Given that this is such a core technique to the paper, this should be described in the manuscript methods.

We have added a detailed description of the AcPIM4 purification and confirmation process to the Methods section (lines 680-700) of the manuscript.

Reviewer #2:

In this work, Liu et al have determined PimE (a mannosyltransferase catalyzing Ac1/2PIM4 and PPP to Ac1/2PIM5 and PP in the periplasmic side of the mycobacterial inner membrane) cryo-EM structures in apo and the product (Ac1/2PIM5 and PP) bound forms, and utilized modeling and molecular dynamics (MD) simulations, as well as extensive functional studies to better understand the PimE mechanism. Compared to NCOMMS-24-53156-T (another manuscript of Liu et al), this manuscript was written much carefully with convincing ample functional studies.

Line 145: There is no Fab in Extended Data Fig. 3a. Should it be Extended Data Fig. 1g?

The correct reference is indeed Extended Data Fig. 1g. We have now corrected this mistake (line 163).

Line 176: It is very difficult to read "yellow" texts in Extended Data Fig. 4.

We appreciate this feedback, and in response have revised Extended Data Fig. 4 accordingly.

Line 187: Fig. 2d-e -> Fig. 2d-e and Extended Data Fig. 5; In the caption of Fig. 2, (e) and (f) need to be swapped.

We have made the following corrections:

1. Correctly referenced both Fig. 2d-e and Extended Data Fig. 5 (now in line 212).
2. The caption for Fig. 2 has been amended, swapping (e) and (f), as suggested (line 924-925).

Figure 4: For both 4e and f, would it be better to have the same orientation as in Fig. 2c and add "CE1" and "CE2" in the figure?

We agree with this suggestion, and in response have revised Fig. 4e and 4f to match the orientation in Fig. 2c. We have also added labels for "CE1" and "CE2" to these panels to facilitate comparisons.

The RoseTTAFold models: It is not clear described how the authors obtained these models. Which templates were used? It is really surprising to get the similar key interactions between PimE and products, compared to the cryo-EM structure. What additional information the authors got from these models?

We thank the Reviewer for raising this point. The RoseTTAFold models provide valuable complementary information to our cryo-EM structures, particularly in visualizing interactions with the full-length substrates where density was not fully resolved in our maps. These models support our experimentally-observed interactions and revealed additional substrate-coordination networks. We have updated our Results section, "Structural and computational insights into ligand interactions," (lines 295-308) to better highlight these points. While RoseTTAFold All-atom uses PDB coordinates for training data, it does not require users to input explicit templates to generate the results shown here.

Extended Data Fig. 8 and Line 294: 1. Can a ligand RMSD of 0.6 nm be considered stable? 2. How a ligand RMSD can be indicative of ligand affinity?

In this case, our interpretation suggests that with an RMSD value of 0.6 nm the ligand remains within the binding site, exhibiting flexibility primarily in its lipid tails and, to a lesser extent, in its sugar headgroups. This stability within the binding site is further supported by the persistent contacts between the ligand and the protein (Extended Data Fig. 8b). Lines 325-326 has been revised for improved clarity.

Extended Data Fig. 9 and Line 309: The RMSF plot should be twice wider in X. More importantly, how the calculated pKa values of D58 and K195 can suggest a possible role in catalysis?

We have included a revised RMSF plot that is twice as wide for better clarity. The range of calculated pK_a values for the product state differs from those of the apo and substrate states for both residues. For D58, the calculated pK_a is consistently higher than the expected value (4.5, indicated by the gray dashed line), suggesting a greater likelihood of protonation at physiological pH. In contrast, the lower pK_a value of K195 compared to the expected value may indicate its role in interacting with D58 or the phosphate group of PPM during the SN2 reaction.

Reviewer #3:

In this manuscript, Liu et al. present single particle cryo-EM structures of the mycobacterial mannosyltransferase PimE in apo and product-bound states. The authors also thoroughly investigate residues involved in catalysis and substrate/product binding using in vitro and in vivo functional assays and molecular dynamics simulations. Overall, the study is well-executed, well-presented and provides important mechanistic insights into the function of PimE and mycobacterial cell envelope biogenesis that will be of interest to a broad scientific audience. I only have minor issues listed below.

We thank the Reviewer for their positive assessment of our work and appreciate their constructive feedback.

Comments regarding the structures:

1. The details for cryo-EM data collection described in the methods and shown in Table 1 are not consistent. Please check.

We apologize for the inconsistency. We have reviewed and corrected the data collection details in both the Methods section (lines 594-620) and Table 1 to ensure they are consistent and accurate.

2. The particle distribution plots in ED Figure 2 indicate potential preferred orientation problems. Please provide 3DFSC curves.

We thank the Reviewer for this observation. In response, we performed 3DFSC analysis for both apo and product-bound PimE reconstructions. The results demonstrate high sphericity values of 0.950 for apo PimE and 0.845 for product-bound PimE, with directional FSC histograms showing a relatively uniform distribution of resolution across different directions. These analyses indicate that any potential preferred orientation does not appear to compromise the quality of the 3D reconstructions.

While ED Fig. 2 shows a higher prevalence of certain views, this apparent orientation bias actually proved advantageous for our structure determination. The dominant side views of the particles

provided excellent visualization of Fab features, as evidenced by the 2D classification averages shown in Extended Data Fig. 2, which served as crucial fiducial markers for efficient particle alignment and reconstruction.

To ensure comprehensive sampling of Fourier space, we collected a large dataset. The quality of sampling is validated by the Sampling Compensation Factor (SCF) values generated in CryoSPARC, which are 0.822 and 0.823 for apo PimE and product-bound PimE, respectively. These values confirm sufficient sampling distributions despite the observed preferred orientation. Additionally, histograms of directional FSC values further illustrate consistent resolution across all directions. We have included these additional analyses in Supplementary Fig. 2.

3. The authors do not present a structural comparison between the apo and product bound structures. What is the C α RMSD, and does this say anything about the mechanism?

We appreciate this suggestion. In our previously submitted manuscript, we did include a structural comparison between the apo and product bound structures (Extended Figure 6d and lines 222-235 in previous manuscript), but in response have now performed a more detailed comparison analysis. The superposition of the apo and product-bound structures reveals close structural similarity, with a C α RMSD of 1.5 Å across all 367 matched C α pairs. This similarity underscores the conserved architecture of PimE between the two states, while the subtle conformational rearrangements, particularly in the peripheral loops (PLs), suggest localized adjustments that may facilitate substrate binding and stabilization. We have updated the Results section titled “The cryo-EM structure of reaction products-bound *Ma*PimE”, accordingly (lines 251-267).

4. Why was the Fab not modelled in the product-bound structure?

The Fab was indeed present in both the apo and product-bound samples. However, in the product-bound structure, the density for the Fab was less well-defined. We have now also deposited the Fab-bound model.

5. Have the authors tried using classification with a mask focusing on the substrate/product cavity to discard any potential particle populations without bound products? This might improve the density of the products.

We appreciate this suggestion. We have indeed tried focused 3D classification using a mask focusing on the substrate-binding cavity. Unfortunately, we did not observe a significant improvement in the density of the bound products or the overall resolution of the binding site.

6. The reaction products are “stuck” on PimE in the structure—how do the authors envisage that the products are released from PimE in vivo, i.e. how is enzyme inhibition by the products prevented?

It is conceivable that sample preparation for structural studies was conducted under conditions that slowed the enzymatic reaction or altered substrate release to some extent, resulting in the capture of a product-bound state of PimE. Alternatively, this could represent a rate-limiting step in the overall production of higher-order PIMs. The nature of the contacts between the substrates and the protein is inherently reflected in the products. In our CG simulations, the observed residence time is approximately 1 μ s.

*7. The authors frame the article by highlighting that PimE could be an antibiotic target against *M. tuberculosis*. Can the authors provide a comparison of their structures with a predicted model of*

M. tuberculosis PimE? Are all important residues identified in this study present in the *M. tuberculosis* homologue?

We thank the Reviewer for this suggestion. To address this, we generated AlphaFold models for both *M. tuberculosis* and *M. smegmatis* PimE and compared them with our *M. abscessus* PimE structure. These comparisons reveal a high degree of structural similarity between the enzymes. Specifically, the identity between *M. abscessus* and *M. tuberculosis* PimE is 64%, with a C α RMSD of 2.5 Å across all 369 pairs, while the identity between *M. abscessus* and *M. smegmatis* PimE is 61%, with a C α RMSD of 1.9 Å across all 369 pairs.

All the residues in *M. abscessus* PimE flagged as important are conserved in both *M. tuberculosis* and *M. smegmatis* PimE, supporting the functional relevance of our findings across these species. We have added a new figure showing the structural alignment and comparison between *M. abscessus*, *M. tuberculosis*, and *M. smegmatis* PimE and updated the related Results section (line 190-201) accordingly.

8. Can the authors provide the Z-scores for the structural homologues identified in the DALI search?

We have now included the Z-scores and relevant statistics for the structural homologues identified in the DALI search in the legend for Extended Fig. 4 (line 1017-1022).

Other comments:

9. Have the authors checked if the Fab used for facilitating structure determination has any effect on the catalytic activity? This might tell something about the catalytic mechanism, e.g. if any conformational changes are required.

The Fab fragment binds to a region of PimE that is distant from the active site and substrate-binding sites. There appears to be no overlap between the Fab-binding epitope and the catalytic or substrate-binding residues, suggesting that the Fab is unlikely to interfere with enzymatic activity.

10. The metal-dependence assay is not comprehensive and relies on chelators to strip PimE of any metals acquired from the expression host rather than adding individual metals and looking for an increase in activity. PimE is clearly still active in the presence of chelators, as shown in ED Figure 1b, but the assay doesn't tell anything about the reaction rate, which might be stimulated by a metal ion. Considering available structural information on metal-dependent glycosyltransferases, is there any putative metal binding site(s) near D58?

We appreciate the Reviewer's comments. As discussed above, to address these concerns, we performed additional activity assays to investigate the potential role of metal ions in PimE function. These experiments included reactions supplemented with Mg²⁺, a common cofactor for glycosyltransferases. The results, presented in Extended Data Fig. 1b (right), show that the addition of Mg²⁺ had no effect on product formation, consistent with our earlier observation that PimE remains active in the presence of chelators (Extended Data Fig. 1b, left). Our structural data do not reveal any density for metal ions near D58 or other catalytically important regions. Moreover, PimE lacks the Dx₂D motif typically associated with metal-dependent glycosyltransferases, again consistent with our observations of metal-independent activity. This finding of metal-independent function of PimE aligns with previous studies of structurally related enzymes ALG6 and AftA, which also lack the Dx₂D motif and operate without metal ions. We have updated the Results (line 370-376) and Discussion (line 435-439) sections of the manuscript to incorporate these findings.

11. The authors suggest that PimE is a potential antibiotic target, but a previous study (PMID: 16803893) showed that a pimE deletion mutant is still viable and has an intact cell wall. Is there any additional evidence in the literature that PimE is a promising antibiotic target?

We appreciate the Reviewer's comment and in response would like to clarify that we did not specifically suggest PimE as an antibiotic target. Rather, our manuscript discusses its potential as a target for anti-TB therapeutics more broadly. While *pimE* deletion mutants are viable (PMID: 16803893), recent studies have shown that PimE disruption leads to increased sensitivity to multiple antibiotics and copper (Eagen et al., 2018), as well as compromised envelope permeability (Xu et al., 2017). We have revised the Introduction (lines 102-106) and Discussion (453-455) sections to clarify this.

12. The authors highlight the structural similarity of PimE and the human GPI transamidase complex, but also suggest that PimE is a good antibiotic target. These statements seem to be at odds—given that both enzymes catalyse reactions involving phosphatidylinositol mannosides, any PimE-targeting antibiotic might also inhibit the human enzyme. Can the authors also clarify how PimE is different to the GPI transaminase?

While PimE and the human GPI transamidase complex share similar structural folds, their active sites and substrate binding pockets differ significantly. We have revised the text to clarify this point, and avoid any potential confusion (line 185-189).

13. L937 – please use either the single- or three-letter amino acid code consistently.

We have checked for consistency throughout the text.

Response to Reviewers

Liu *et al.* “Mechanistic studies of mycobacterial glycolipid biosynthesis by the mannosyltransferase PimE”.

Reviewer #1:

The authors have addressed all of my concerns. I want to congratulate them on a very nice paper.

Reviewer #2:

The authors have addressed all my comments and questions properly.

Reviewer #3:

Thanks to the authors for positively addressing my and other reviewers' comments. The clarity of the manuscript is now much improved and I have no further issues.

We are grateful to all the Reviewers for their appreciation of our work and thank them again for their constructive criticisms that have been truly important for us to be able to improve the manuscript.